# Vertical distribution of the particle phase in tropical deep-convective clouds as derived from cloud-side reflected solar radiation measurements

Evelyn Jäkel[1], Manfred Wendisch[1], Trismono C. Krisna[1], Florian Ewald[2,3], Tobias Kölling[2], Tina Jurkat[3], Christiane Voigt[3], Micael A. Cecchini[4], Luiz A. T. Machado[4], Armin Afchine[5], Anja Costa[5], Martina Krämer[5], Meinrat O. Andreae[6,7], Ulrich Pöschl[6], Daniel Rosenfeld[8], and Tianle Yuan[9]

[1]Leipzig Institute for Meteorology (LIM), University of Leipzig, Germany.
[2]Meteorological Institute, Ludwig-Maximilians-University Munich, Germany.
[3]Institut für Physik der Atmosphäre, Deutsches Zentrum für Luft und Raumfahrt (DLR), Oberpfaffenhofen, Germany.
[4]Center of Weather Forecast and Climates Studies (CPTEC), National Institute for Space Research (INPE), Sao Jose Dos Campos, Brazil.
[5]Forschungszentrum Jülich, Jülich, Germany.
[6]Max Planck Institute for Chemistry (MPIC), Mainz, Germany.
[7]Scripps institution of Oceanography, University of California San Diego, La Jolla, California, USA.
[8]Institute of Earth Sciences, The Hebrew University of Jerusalem, Jerusalem, Israel.
[9]NASA Goddard Space Flight Center, Greenbelt, Maryland, USA.

*Correspondence to:* E. Jäkel
e.jaekel@uni-leipzig.de

**Abstract.** Vertical profiles of the cloud particle phase state in tropical deep-convective clouds (DCCs) were investigated using airborne solar spectral radiation data collected by the German High Altitude and Long Range Research Aircraft (HALO) during the ACRIDICON-CHUVA campaign, which was conducted over the Brazilian rainforest in September 2014. A phase discrimination retrieval based on imaging spectroradiometer measurements of DCC side spectral reflectivity was applied to clouds formed in different aerosol conditions. From the retrieval results the height of the mixed phase layer of the DCCs was determined. The retrieved profiles were compared with in situ measurements and satellite observations. It was found that the depth and vertical position of the mixed phase layer can vary up to 900 m for one single cloud scene. This variability is attributed to the different stages of cloud development in a scene. Clouds of mature or decaying stage are affected by falling ice particles resulting in lower levels of fully glaciated cloud layers compared to growing clouds. Comparing polluted and moderate aerosol conditions revealed a shift of the lower boundary of the mixed phase layer from 5.6±0.2 km (269 K) [moderate] to 6.2±0.3 km (267 K) [polluted], and of the upper boundary from 6.8±0.2 km (263 K) [moderate] to 7.4±0.4 km (259 K) [polluted], as would be expected from theory.

# 1 Introduction

Deep-convective clouds (DCCs) play a crucial role in redistributing latent heat, influencing the hydrological cycle, and regulating the radiative energy budget of the Earth's climate system. In particular, tropical convection is a key component of the global circulation of the atmosphere, which is the primary pathway for energy transport from the tropics to the mid-latitudes. DCCs exhibit a high variability of cloud particle sizes and a complex vertical microphysical structure. This includes the different phase states of water (liquid and ice) of the cloud particles and the occurrence of layers where phase transitions between liquid water and ice particles (further referred to as mixed phase) take place. The optical, microphysical, and macrophysical properties of DCCs determine their radiative effects and are controlled by particle growth occurring within the clouds. Consequently, the understanding of the processes driving the evolution of DCCs is of major importance. In particular, aerosol particles modify cloud properties, including their radiative effects (Twomey, 1977), as well as their lifetime and the formation of precipitation (Albrecht, 1989). Many efforts have been undertaken to quantify these effects, which take place over a wide range of spatial and temporal scales (Rosenfeld et al., 2014). Aerosol particles have an influence on the cloud droplet size distributions (more aerosol particles lead to more and smaller cloud droplets), on warm rain and cold rain development, on the cloud top height evolution, the depth of the mixed phase layer, and the occurrence of lightning (Tao et al., 2012). While the formation of warm rain is suppressed by enhanced aerosol particle number concentration, the cold-rain evolution is intensified due to extra latent heat, which leads to an invigoration of the DCC development (Andreae et al., 2004; Rosenfeld et al., 2008). The phase transition from liquid water to ice is especially relevant for the development of precipitation. Furthermore, the optical properties of ice and liquid water clouds differ and, thus, cause variable radiative effects. Rosenfeld and Lensky (1998) found that in continental clouds glaciation occurs at much colder temperatures ($-15°$ C to $-30°$ C) than in maritime clouds (warmer than $-10°$ C). Consequently, the vertical transitional mixed phase zone in continental clouds is geometrically thicker than in maritime clouds. In polluted clouds the coalescence zone vanishes (in which droplet growth by collision and coalescence play a major role), and mostly small liquid water droplets are observed. The mixed phase zone is shifted to lower temperatures (less than $-15°$C), and glaciation occurs often above the -30°C isotherm, with the extreme situation of polluted clouds with strong updrafts reaching to -38°C (Rosenfeld and Woodley, 2000).

Profile measurements of microphysical structure and formation of precipitation remain a challenge. Either in situ measurements (Freud et al., 2008; Konwar et al., 2012; Khain et al., 2013, e.g.,) or remote sensing techniques are applied to obtain profiles of cloud microphysical parameters, such as cloud particle size and phase state. Active remote sensing observations (e.g., radar) provide profiles along the line-of-sight. These sensors may penetrate through a cloud, but the quantitative retrieval of cloud optical and microphysical properties is problematic since the signal is dominated by scattering due to large droplets.

Rosenfeld and Lensky (1998) introduced a method to derive vertical profiles of the effective droplet radius as a function of brightness temperature from satellite reflectance measurements. They analyzed clusters of convective clouds at different stages of vertical development to retrieve the temporal evolution of individual cloud elements. This ensemble method assumes that cloud-top properties derived from clouds at different stages of their evolution are comparable to the properties of an individual cloud as it evolves through the various heights (Freud et al., 2008). From the ensemble of retrieved effective droplet sizes, a

vertical profile of cloud phase can be estimated because of the relationship between cloud phase and vertical profile of the cloud particle size (Rosenfeld and Feingold, 2003; Yuan et al., 2010; Martins et al., 2011). However, the retrieval of the effective droplet size relies on one-dimensional (1D) radiative transfer simulations, which incorporates retrieval uncertainties due to plane-parallel cloud assumptions and neglecting the net horizontal radiative transport between the satellite pixels (Zinner et al.,

2006). Consequently, a decrease of pixel size causes an increase of the independent pixel bias, because the smaller the pixel, the more important is the net horizontal photon transport, particularly for the wavelengths in the visible spectral range, which are used for the retrieval of the effective droplet radius.

The retrieval uncertainty due to the 1D approximation and the assumptions made with respect to the ensemble method can be mitigated by using multi-angle spectroradiometer measurements (ground-based, airborne, or satellite) of cloud side spectral

reflectivity. A step further is the application of high-resolution imaging spectroradiometers, which enables profiling of individual clouds with a temporal resolution of one minute from both ground or aircraft. For airborne applications there are no safety-related flight restrictions due to strong turbulences and icing as would be required in case of cloud penetrations for in situ probing.

The retrieval approach of the thermodynamic water phase based on cloud side observations exploits the differences in the imag-

inary part of the refractive index of the cloud particles of both phases in the near infrared (NIR: 0.7-2.5 $\mu$m) wavelength range (Pilewskie and Twomey, 1987; Ehrlich et al., 2008; Martins et al., 2011; Jäkel et al., 2013). While Pilewskie and Twomey (1987) and Jäkel et al. (2013) applied ground-based measurements of spectral reflectivity between $1.5 - 1.7$ $\mu$m wavelength for the phase discrimination, Martins et al. (2011) and Marshak et al. (2006) utilized reflected radiation data at 2.10 $\mu$m and 2.25 $\mu$m wavelength. A phase index was defined by Jäkel et al. (2013) using the spectral slope of cloud side reflected radiances between

1.55 and 1.7 $\mu$m. Jäkel et al. (2013) showed by applying three-dimensional (3D) radiative transfer simulations that this slope is negative for liquid water and positive for ice particles, mostly independent of the viewing geometry and cloud particles size. For DCCs with liquid water, ice particles, and mixed phase layers, profile measurements of the phase index provide evidence where and in which stage of development ice particles start to form. For ground-based observations, Jäkel et al. (2013) identified the mixed phase zone by a strong increase of the phase index from negative to positive values, while the vertical profile of

the phase index for pure liquid water or ice particles is less variable.

To determine the height and temperature of the mixed phase layer from cloud side spectral reflectivity observations additional information is required. Martins et al. (2011) used a thermal infrared sensor at 11 $\mu$m wavelength yielding the brightness temperature, which is an indicator of cloud height. Collocated scanning active remote sensing techniques by radar or lidar were applied to estimate geometric information on cloud distance and height (Jäkel et al., 2013; Ewald et al., 2015). Another method

is based on stereographic analysis of multiangle observations (e.g., Seiz and Davies, 2006).

Different from the scanning-point-sensor measurements as presented by Martins et al. (2011), this paper introduces airborne measurements of an imaging spectroradiometer called specMACS (spectrometer of the Munich Aerosol Cloud Scanner, Ewald et al., 2016). These observations were used to derive vertical profiles of the phase state of DCCs during the HALO (High Altitude and Long Range Research Aircraft) campaign ACRIDICON (Aerosol, Cloud, Precipitation, and Radiation Interac-

tions and Dynamics of Convective Cloud Systems) - CHUVA (Cloud processes of tHe main precipitation systems in Brazil:

A contribUtion to cloud resolVing modeling and to the GPM (GlobAl Precipitation Measurement)) in 2014 (Wendisch et al., 2016). The measurement technique of imaging spectroradiometers allows instantaneous spectral cloud side observations for a set of viewing angles depending on the number of spatial pixels of the sensor. The imaging spectroradiometer measurements were supplemented by video camera observations to estimate the cloud distance and height from stereographic analysis.

In this paper we will address the following questions: (i) Can we observe differences in the vertical distribution of the thermodynamic phase state in DCCs for different aerosol conditions by using cloud side observations? (ii) How do the vertical profiles of cloud phase derived from cloud side observations agree with results from satellite (ensemble method) and in situ measurements?

    The instrumentation and the field campaign are introduced in Section 1, followed by a description of the methodology of

the phase retrieval (Section 2). In Section 3 the method is applied to data from three flights conducted during ACRIDICON-CHUVA. The variability of vertical phase distribution is discussed with respect to aerosol conditions and compared to in situ and satellite products.

## 2   Measurements and tools

### 2.1   Field campaign

Airborne remote sensing and in situ data sampled during ACRIDICON-CHUVA are used to derive vertical profiles of the thermodynamic phase (ice or lqiud water) of cloud particles in DCCs as measured over the Brazilian rainforest. Local convection is strongly forced by the diurnal cycle. In particular, at the end of the dry season (September), a large variability of aerosol particles due to biomass burning is observed (Andreae et al., 2015). Three out of fourteen scientific flights (labelled as AC10, AC13, AC18) are selected for this study (flight tracks shown in Fig. 1) covering an area of about $1400 \times 1200$ km$^2$.

The temperature profiles of the three flights show only small day–to–day variations in spite of the different flight directions. In contrast, the relative humidity is variable with flight area and altitude as was shown by Cecchini et al. (2017b). They discussed in particular the relation between cloud base and humidity below clouds for several flights performed during the ACRIDICON-CHUVA campaign. For AC13 they found less relative humidity (75 %) and a higher cloud base (2000 m) due to deforestation than compared to measurements over the rain forest (80 % relative humidity and 1500 m cloud base). In the overview paper of

the ACRIDICON-CHUVA campaign by Wendisch et al. (2016) the aerosol conditions from AC13 was classified as polluted. Cecchini et al. (2017a) used the aerosol concentration measured with a condensation particle counter (CPC) at cloud base for flights AC13 and AC18 as indicator. They found 4100 particles cm$^{-3}$ for AC13 suggesting polluted clouds and about 740 particles cm$^{-3}$ for AC18 indicating clouds under Amazonian background conditions typical of the dry season. No appropriate measurements at cloud base are available for AC10. Ground-based measurements on this day at the Amazonian Tall Tower

Observatory (ATTO) located at -2.143°S and -59.001°W revealed a particle concentration between 1100 and 1600 cm$^{-3}$. Since the flight AC10 was in the same general region, these data are used to describe the aerosol condition of AC10. Furthermore, the aerosol optical depth in the main measurement areas taken from MODIS (Moderate-resolution Imaging Spectroradiometer) product MOD04/MYD04 (3-km-pixel resolution) are chosen as additional parameter. Quite variable values between 0.3-0.4

for AC18 (28 September 2014), between 0.4-0.5 for AC10 (12 September 2014) and between 0.5-0.6 for AC13 (19 September 2014) are found. From these data AC10 and AC18 are classified as moderate aerosol cases. A summary of the three flights used in this work is given in Table 1.

## 2.2 Instrumentation

### 2.2.1 specMACS and GoPro

The imaging spectroradiometer specMACS (Ewald et al., 2016) consists of two line cameras (manufactured by SPECIM, Finland), one for the visible and near-infrared (VNIR), the other for the shortwave infrared (SWIR) spectral range. The field of view (FOV) along the spatial lines of both cameras differs slightly ($33°$ and $35°$) due to different optics. The incoming solar radiation is distributed over 1312 and 320 spatial pixels. For each spatial pixel, spectral information can be measured within $0.4 - 1.0$ $\mu$m (800 spectral channels) and $1.0 - 2.5$ $\mu$m (256 spectral channels), with a bandwidth between $2.5 - 12.0$ nm. SpecMACS was characterized in the laboratory with respect to nonlinearity, dark current, and polarization (Ewald et al., 2016). Spatial calibrations were performed to derive the angular resolution of both sensors, which is needed for final geometric matching of both sensors. The spectral characteristics were deduced by using monochromator output at selected wavelengths. The absolute radiometric response was determined using an integrating sphere and the absolute RAdiance STAndard (RASTA; Schwarzmaier et al., 2012) traceable to absolute radiance standards of PTB (Physical Technical Bundesanstalt). The wavelength-dependent uncertainties ($2\sigma$) of the absolute radiometric response including sensor noise and dark current drift between 3 % and 14 % (in the outer region of the measured spectra) were given in Ewald et al. (2016).

During the ACRIDICON-CHUVA campaign, specMACS was mounted at a side view port on HALO. The transmission of the optical window with purified quartz glass panes (type: Herasil 102) was characterized in the laboratory. The line cameras were orientated in vertical position as illustrated in Fig. 2. During the aircraft movement 3D (two spatial, one spectral dimension) snapshots of cloud scenes were taken.

For estimates of the cloud distance a two-dimensional (2D) digital action camera (type: Hero HD3+ 3660-023 Full-HD manufactured by GoPro, Inc., USA, and hereafter called GoPro) was installed at the side window of HALO. Movies with full HD at a resolution of $1920 \times 1080$ pixels were recorded during the flight. The original lens of the camera was replaced by a distortion free optics covering a horizontal $FOV_h$ of about $90°$ and a vertical $FOV_v$ of about $59°$. A schematic of the setup is shown in Fig. 2. The geometrical calibration of the camera was performed using a square chessboard. Images from different perspectives of the chessboard were taken and evaluated by an open source routine (http://opencv.org) implemented in computer vision algorithms (Bradski and Kaehler, 2013). This allows assigning elevation and azimuthal angle to each point of the image.

### 2.2.2 NIXE-CAPS

In situ measurements of the asphericity of particles were performed with the Novel Ice eXpEriment – Cloud, Aerosol and Precipitation Spectrometer (NIXE-CAPS). The instrument is a combination of two probes, the NIXE-CAS (Cloud and Aerosol Spectrometer) and the NIXE-CIP (Cloud Imaging Probe). While the NIXE-CIP detects the size of particles between 15 and 900

$\mu$m by recording 2-D shadow cast images, the NIXE-CAS measures the size and asphericity of the particles for a range of 0.6 and 50 $\mu$m (Meyer, 2012; Luebke et al., 2016; Costa et al., 2017). NIXE-CAS discriminates between spherical and aspherical particles by measuring the change of the polarized components of the scattered laser light in the backward direction, which is sensitive to the particle shape. Spherical particles are not supposed to alter the polarization state of the incident light as discussed by (Meyer, 2012), while non-spherical ice crystals change the polarization depending on their size and orientation (Nicolet et al., 2007; Meyer, 2012). With respect to the phase state discrimination, aspherical particles can be considered as ice particles. In contrast, spherical particles indicate mainly liquid droplets. Note, that while Järvinen et al. (2016) have shown that ice particles can also be spherical, the large majority of spherical particles is associated with the liquid phase. The ACRIDCON-CHUVA data set is classified with respect to temperature, asphericity, and particle number concentration as measured by NIXE-CAPS (see Table 2).

### 2.2.3 CAS-DPOL and LWC hotwire

The CAS-DPOL (Cloud and Aerosol Spectrometer, with detector for polarization) instrument measures aerosol and cloud particles in the size range between 0.5 and 50.0 $\mu$m (Braga et al., 2016; Voigt et al., 2017) by sensing individual particles passing a focused laser beam. The resulting intensity distribution of forward and backward scattered light is used to derive the size distribution of the particles. Only particles with diameters between 3 and 50 $\mu$m and with a total number density larger than 1 cm$^{-3}$ are classified. Additionally, CAS-DPOL is used to estimate the phase of the cloud particles (liquid or ice). The aspherical fraction (AF) from the CAS-DPOL is determined by measuring the perpendicularly polarized light in the backward direction and the forward scattering light intensity. The ratio of the forward and the backward scattered light determines the phase of the particle. Particles with a polarization ratio larger than the 1-$\sigma$ range of the inferred sphericity- threshold are categorized as aspherical. The method gives a size dependent aspherical fraction of the first 300 particles measured each second. The bulk aspherical fraction is derived from the number of aspherical particles to the number of total particles measured between 3 and 50 $\mu$m per second. Calibration of the backward channel was performed during RICE03 (Rough ICE campaign) at the AIDA (Aerosol Interactions and Dynamics in the Atmosphere) cloud chamber (Järvinen et al., 2016; Schnaiter et al., 2016). Spherical liquid particles reveal a low AF ($< 0.1$) while aspherical particles (ice or aerosols) have a high AF ($> 0.1$, mean of 0.4). Aspherical ice particles may have an AF $< 1$ since the orientation of the particles in the sampling volume may appear circular.

The liquid water content (LWC) was measured with a King type LWC Hotwire (Braga et al., 2016) installed on the CAS-DPOL. The Hotwire sometimes returns a signal in ice or clouds of partly frozen particles. This signal is on the order of 0.2 g m$^{-3}$. Thus, a conservative threshold of 0.3 g m$^{-3}$ is used to reduce the false alarm rate.

### 2.2.4 MODIS

MODIS cloud products (Collection 6) of the Terra (MOD06) and Aqua (MYD06) satellites are used for a comparison of the phase state and glaciation temperature. Since MODIS mainly measures cloud top properties, the time–space–exchangeability of convective clouds as proposed by Rosenfeld and Lensky (1998) is applied and referred to as ensemble method. The cloud

particle phase of the cloud tops is directly taken from the MOD06/MYD06 product "Cloud_Phase_Infrared" with a 1-km-pixel resolution (Baum et al., 2012). Compared to Collection 5, where the cloud phase product was classified as ice, liquid water, mixed phase, and uncertain using brightness temperatures measured at 8.5 and 11 $\mu$m (Platnick et al., 2003), Collection 6 is modified by using additional cloud emissivity ratios (7.3/11, 8.5/11, and 11/12 $\mu$m) as reported by Pavolonis (2010) and Baum et al. (2012). Empirically derived thresholds of these emissivity ratios were defined to separate finally between liquid water and ice clouds. Note, that due to several ambiguities (see Platnick et al. (2017)) a separate classification of mixed phase cloud pixels is no longer provided in Collection 6. The "mixed phase" and "uncertain" classes from Collection 5 are now combined into a single class specified as "undetermined". Hence, the description of the cloud phase profile by applying the ensemble method on the "Cloud_Phase_Infrared" product is limited to the liquid water and the ice phase distribution.

Therefore, the cloud particle size product is used additionally to estimate the glaciation temperature as proposed by Yuan et al. (2010). The vertical distribution and evolution of cloud particle size inside a DCC provides useful information on the phase state (Rosenfeld and Feingold, 2003). The mixed phase layer is characterized by a strong increase of cloud particle size with height (Martins et al., 2011), whereas for fully glaciated cloud layers the largest ice particles can be found directly at the height where the glaciation temperature is reached. At lower temperatures, no supercooled droplets are left for particle growth and only small ice particles are able to move upward inside weakened updrafts. Consequently, the height and temperature where the increase of particle size turns into a decrease is considered as glaciation level and temperature. A sufficiently large statistics is required for the ensemble method. The cloud particle sizes from the MOD06/MYD06 product are averaged for a bin of cloud brightness temperatures (Channel 31; 11 $\mu$m). In contrast to the original retrieval (Yuan et al., 2010), the restrictions concerning cloud optical depth (COD > 30) and cloud top temperature (CTT < 260 K) were relaxed to COD > 10 and CTT < 280 K, to enlarge the statistics of the data.

## 2.3   Radiative transfer model

3D radiative transfer modeling is performed with the forward-propagating Monte Carlo photon-transport model MCARATS (Monte Carlo Atmospheric Radiative Transfer Simulator) (Iwabuchi, 2006). The optical properties (single scattering albedo, extinction coefficient, and phase function) of atmospheric components are pre-defined for each grid cell of the model domain as either horizontally inhomogeneous or homogeneous layers. For the model input, the atmospheric profiles of temperature, atmospheric pressure, and gas densities are taken from Anderson et al. (1986). From a radio sounding from Alta Floresta (-9.866°S, -56.105°W) and measurements of temperature, humidity and pressure performed by HALO, the temperature and pressure profiles are adjusted to represent the atmospheric conditions on 19 September 2014 (AC13) in the region of one of the measurement flights (representative of the three flights considered in this study). The density of water vapor is re-calculated using the relative humidity, temperature and pressure measurements. Since Rayleigh scattering is calculated from the density profile according to Bodhaine et al. (1999), the LOWTRAN (Low Resolution Transmission Model) parametrization by Pierluissi and Peng (1985), as adapted from SBDART (Santa Barbara DISORT Atmospheric Radiative Transfer) (Ricchiazzi and Gautier, 1998) is used for gas absorption. The optical properties of clouds are derived from profiles of effective radius ($r_{\mathrm{eff}}$) and liquid (ice) water contents (LWC, IWC) using Mie calculations for water clouds, while for ice clouds the parameterizations by Baum et al.

(2005, 2007) are used. For the polluted case, aerosol properties are described with the model by Shettle (1989) and scaled by AERONET (AErosol RObotic NETwork) measurements (site Alta Floresta) of aerosol optical depth, single scattering albedo, and asymmetry parameter (used for the Henyey-Greenstein phase function).

## 3  Methodology

The retrieval method of the phase state consists of three main steps: (3.1) The cloud masking procedure to filter illuminated cloud regions, (3.2) the cloud phase discrimination, and (3.3) the geometric allocation of the classified cloud profiles with respect to height and temperature.

### 3.1  Cloud masking procedure

Compared to illuminated cloud sides, the photon paths in shadowed cloud regions are longer, which is related to more absorption events. This absorption due to cloud particles is not locally restricted to the cloud side parts where the camera is pointed at. In fact, the spectral radiation coming from shadowed cloud regions is affected by absorption by cloud particles from cloud parts outside the FOV of each individual spatial camera pixel. Since the spectral signature of reflected radiation from shadowed regions of cloud sides is contaminated by a significant fraction of diffuse radiation originating from unknown cloud regions, a cloud masking technique was developed to discriminate illuminated and shadowed cloud regions. In ground-based observations the reflected radiation measured from shadowed cloud regions showed spectral signatures influenced by the spectral surface albedo due to interaction between clouds and the surface (Jäkel et al., 2013). This interaction is reduced for several reasons for aircraft observation of DCC. The reflected radiation is observed from higher altitudes than from the ground. This is related to changes of the range of scattering angles. Furthermore, the distances between surface and in particular the upper parts of the clouds are much larger. Therefore, scattered radiation from the immediately adjacent cloud regions has a greater effect on the spectral features in the shadowed cloud areas than the surface. Since spectral indication of the surface could neither be observed nor simulated for airborne measurements, a different approach is chosen based on the distribution of color values in the observed cloud scene. Three wavelengths ($\lambda_{\mathrm{B}} = 436$ nm, $\lambda_{\mathrm{G}} = 555$ nm, and $\lambda_{\mathrm{R}} = 700$ nm) corresponding to wavelengths of the RGB (Red Green Blue) color space are selected to calculate a simplified RGB color value for each measured spectrum, which takes into account the sensitivity of the human eye on the different colors by differential weighting of the three wavelengths (IEC, 1999):

$$\mathrm{RGB} = 0.2126 \cdot \mathrm{R} + 0.7152 \cdot \mathrm{G} + 0.0722 \cdot \mathrm{B} \tag{1}$$

where R, G, and B represent the normalized spectral radiances. The histogram of the RGB color values for each cloud scene is used to identify the illuminated and shadowed cloud areas.

Before showing an application, the procedure is illustrated using simulated cloud side reflectivity observations. In this manner, we can directly compare the classification of illuminated and shadowed cloud regions (i) derived from known cloud and viewing geometry, and (ii) derived from the histogram of the RGB color values. The cloud field was generated by the Goddard Cumulus

Ensemble model (Tao et al., 2003; Zinner et al., 2008) for a model domain of $64 \times 64$ km with a horizontal resolution of 250 m and a vertical resolution between 0 and 10 km altitude of 200 m. From 10 to 120 km altitude the simulations are performed with a vertical resolution ranging between 1 and 5 km. The maximum extension of the liquid water clouds from bottom to cloud top ranges from 1.0 to 7.4 km altitude. As MCARATS is a forward-propagating radiative transfer model (RTM) the simulations are performed for each grid point representing an observation altitude of 4 km. The sensor is pointed at an elevation angle of $10°$ and with a relative azimuth angle to the Sun of $60°$ to trigger also areas of shadowed clouds. Fig. 3a displays the RGB color values derived from the radiance simulations at each of the $256 \times 256$ grid points. From information of the viewing geometry of the sensor and Sun (solar zenith angle $\theta_0 = 30°$) and the setup of the clouds in the model domain, each observed cloud pixel is classified as shadowed or illuminated. The histogram of the simulated RGB color values is shown in Fig. 3b as black line. Two modes are visible, which coincide with the two sub-classes of illuminated (red) and shadowed (blue) cloud regions which were calculated from the cloud and viewing geometry. To identify the illuminated cloud areas for an unknown cloud geometry, as is the case for real measurements, only the brightest pixels that correspond to the right-most mode in the RGB-histogram are selected. Since the left side of this mode may also include data from shadowed regions, data larger than the maximum of this mode will be classified as illuminated and used for the cloud phase retrieval.

The procedure is applied for an example cloud scene observed during ACRIDICON-CHUVA from 19 September 2014. During the roughly one minute flight leg the aircraft did not change its flight attitude, resulting in almost constant relative azimuth angle (angle between the sun and the viewing direction of specMACS) of $68°$ and solar zenith angle ($\theta_0 = 39°$). Note, that all other selected cloud cases in this study have similar restrictions concerning the flight attitude and time period (about one minute) to guarantee comparable illumination conditions in one cloud scene. Fig. 3c illustrates the RGB histogram as calculated for observations of specMACS with an elevation ranging between -13 and $+12°$. The inlay in Fig. 3c shows the cloud situation as observed from specMACS. Applying the threshold criteria to identify the illuminated cloud parts gives a cloud mask as presented in Fig. 3d, where the illuminated cloud parts are highlighted.

## 3.2 Cloud phase discrimination

Vertical profiles of the relationship between temperature and particle size to identify the mixed phase cloud layer have been used by e.g., Rosenfeld and Woodley (2003). For continental conditions (as often observed in the Amazon Basin) the droplet size may not significantly increase between the main coalescence and mixed phase regions. Therefore, for these cases it is difficult to define the height or temperature where phase transition takes place by the increase of the droplet size. As presented in Ehrlich et al. (2008), Jäkel et al. (2013), and Jäkel et al. (2016), another method based on differences of the refractive index of ice and liquid water between 1550 and 1700 nm wavelength can be applied to discriminate the thermodynamic water phase. The so-called phase index $I_P$ based on spectral radiances ($I$) was introduced as:

$$I_P = \frac{I_{1700} - I_{1550}}{I_{1700}} \qquad .$$

(2)

For ground-based application with corresponding viewing geometry vertical profiles of the phase index were simulated by Jäkel et al. (2013). A significant gradient in the vertical profile of the phase index was observed between liquid water and

mixed phase layer, but also between mixed phase layer and ice phase. A similar behavior was also found for the reflectance ratio at 2.10 and 2.25 $\mu$m as reported by Martins et al. (2011). They observed a strong gradient in the profile of the reflectance ratio. This is due to the fact, that the imaginary part of the refractive index, which determines the spectral absorption, is different between ice and liquid water particles in the two wavelength ranges used by Martins et al. (2011) and Jäkel et al. (2013).

In the following, results from radiative transfer simulations using MCARATS are presented. The viewing geometry and the atmospheric description are adapted to the conditions during ACRIDICON-CHUVA on 19 September 2014. These simulations are performed to demonstrate that ice and liquid water phase can be separated from the transition layer under different conditions similar to the results reported by Jäkel et al. (2013). Note, that due to the different viewing geometry, another angular range of the scattering phase function was observed than for ground-based measurements. This might have an effect on the

characteristics of phase index profile in particular with respect to separation of the mixed phase layer. The model domain used for the simulations had $140 \times 40 \times 99$ grid cells at a horizontal resolution of 250 m and a vertical resolution of 200 m below 14 km altitude and variable resolution above. For each grid cell in a flight altitude of 8 km the spectral radiance at 1550 and 1700 nm wavelength is simulated for sensor viewing elevation angles between -20 and +20° corresponding to the FOV of specMACS. Two simplified cloud scenarios with different profiles of cloud effective radius and water content are assumed. In

both scenarios the clouds ranges from 4.0 to 11.0 km altitude with a mixed phase layer between 6.4 and 7.0 km. While the first scenario uses constant values of cloud effective radius ($r_{\mathrm{eff}} = 20$ $\mu$m for liquid water and ice) and water content (0.7 $\mathrm{gm}^{-3}$), the second scenario assumes variable profiles of the microphysical parameters. These two scenarios are chosen to identify effects on the $I_{\mathrm{P}}$-profile caused by changes of (i) the phase state itself (scenario 1), and (ii) the cloud particle size and water content (scenario 2). From the 3D simulations of the spectral radiance at 1550 and 1700 nm the phase index is calculated following Eq.

(2). For each modeled grid cell in the model domain with a horizontal distance between 3 and 8 km to the cloud, a combined $I_{\mathrm{P}}$-profile is derived from the different viewing elevation angles. Such $I_{\mathrm{P}}$-profiles are plotted in Fig. 4a in black dots. Due to the variation of cloud distance and viewing elevation angle, the $I_{\mathrm{P}}$-profile comprises reflected radiances originating from various scattering angles. For the first scenario with constant microphysical parameters, three distinct clusters corresponding to the phase state of water and the zone of phase transition, with negative values for pure liquid water, can be found. In the mixed

phase layer the phase index shows a steep increase to values larger than 0.15. The absolute difference of the phase indices between mixed phase layer and pure ice phase layer is less pronounced than between liquid and mixed phase layer. This might be caused by the fact that the contribution of ice particles within the mixed phase layer leads to an increased absorption of radiation resulting in an increase of the phase index. The variability of the phase index for constant microphysical conditions in each of the phases is caused by the effect of the different viewing geometries. The vertical cloud structure is observed from dif-

ferent sensor elevation angles and distances. As the scattering phase function depends on the scattering angle, the wavelength and the particle shape, the viewing geometry of the sensor relative to the position of the Sun (here: $\theta_0 = 30°$) also modulate the phase index. The second cloud scenario assumes variable cloud microphysical properties. In general, in convective clouds, the size of ice particles is higher than the size of liquid water particles. Therefore, the second scenario represents a more realistic vertical distribution of the particle effective radius and water content than the first scenario. The corresponding vertical profiles of the effective radius and the water content of the cloud are plotted in Fig. 4b. The mixed phase layer is characterized by the

maximum particle sizes of liquid and ice particles over the entire profile, but lower water content compared to regions above and below. As concluded by Jäkel et al. (2013), the phase index becomes less variable for a water content of more than 0.4 g m$^{-3}$ (variation lower than 7 %). This holds true for most of the DCCs when cloud edges are excluded, which are optically thinner than the inner regions of the cloud. Consequently, mainly the particle size and the phase state drive the changes of the phase index with height. Less impact is attributed to the change of the sensor elevation angle, since the variability of the phase index with respect to the viewing geometry for each phase state in the first cloud scenario with fixed cloud microphysics is lower than the variability of $I_P$ due to the changed cloud properties in the second cloud scenario. The mixed phase layer for the second scenario is characterized by a significant increase of the phase index with height. Once the pure ice phase is reached, the slope of $I_P$ decreases. In the following, the magnitude of vertical change of the phase index will serve as indicator of the position of the mixed phase layer.

## 3.3 Cloud geometry retrieval

Due to the spatial dimension of the specMACS-SWIR instrument, reflected radiances are measured for 320 different angles with an average pixel-to-pixel spacing of about $0.11°$. To quantify the vertical position of the mixed phase layer in terms of height or temperature, information on the cloud distance is required. For that purpose, collocated images of the GoPro camera are combined with flight attitude data to apply stereo-photogrammetric methods. The theoretical background on photogrammetry is given in Hartley and Zisserman (2004), while Hu et al. (2009) applied these techniques for cloud geometrical reconstruction. The mathematics for the geometry retrieval, as it is used in this study, is based mainly on the method described by Biter et al. (1983). They deployed a side-looking camera onboard of an aircraft to detect the position of cloud features, similar to the setup presented in this work.

To estimate the distance to the observed cloud element (C) two images from different positions (P1 and P2) with a projection of the observed point in both images need to be taken (C1 and C2, so-called tie points) as illustrated in Fig. 5a. The geometric problem comprises three coordinate systems: for the camera, the aircraft, and the world coordinate system (longitude, latitude and altitude) for the observed point C (Biter et al., 1983). Coordinate transformations are required to relate the different coordinate systems. Fig. 5b illustrates the aircraft and camera coordinate system, which differ because the GoPro camera looks perpendicular to the flight direction. For example, a positive pitch angle of the aircraft (associated with rotation around the aircraft $y_a$-axis) rotates the camera (image) around the camera's $x_i$-axis as can be deduced from Fig. 5b. The $x$-and $y$-axis of the world coordinate system (not shown) are pointed to the east and to the north, respectively, while the $z$-axis is perpendicular to the $x$-$y$ plane (pointing upward). Each selected image in the camera system ($x_i$, $y_i$, $z_i$) is transformed into the aircraft coordinate system ($x_a$, $y_a$, $z_a$), and finally into the world system ($x_w$, $y_w$, $z_w$). This transformation requires the rotation of the coordinate systems with respect to the three Euler angles pitch, roll, and yaw using the $3 \times 3$ rotation matrices for the aircraft to world $[\mathbf{R}_w^a]$, and camera to aircraft $[\mathbf{R}_a^i]$ system:

$$\begin{bmatrix} x_w \\ y_w \\ z_w \end{bmatrix} = [\mathbf{R}_w^a][\mathbf{R}_a^i] \begin{bmatrix} x_i \\ y_i \\ z_i \end{bmatrix} . \tag{3}$$

The general form of the two rotation matrices for system 1 to system 2 (either "a" to "w" or "i" to "a") are:

$$[\mathbf{R}_2^1] = \begin{bmatrix} \cos\psi\cos\theta & \cos\psi\sin\theta\sin\phi + \sin\psi\cos\phi & -\cos\psi\sin\theta\cos\phi + \sin\psi\sin\phi \\ -\sin\psi\cos\theta & -\sin\psi\sin\theta\sin\phi + \cos\psi\cos\phi & \sin\psi\sin\theta\cos\phi + \cos\psi\sin\phi \\ \sin\theta & \cos\theta\sin\phi & \cos\theta\cos\phi \end{bmatrix} \tag{4}$$

with $\phi = -(\phi_a - 180°)$, $\theta = \theta_a$, and $\psi = (\psi_a - 90°)$ for aircraft to world coordinates and $\phi = -\phi_i$, $\theta = -\theta_i$, and $\psi = -\psi_i$ for camera to aircraft coordinates.

After coordinate transformation, trigonometric methods (Biter et al., 1983) are applied to calculate the distance between the

camera positions P1 and P2 to the observed point C. Repeating this procedure for a number of points yields a relation between elevation angle and cloud height. Note, that the elevation angle represents the elevation angle of the selected tie point of the camera image after correction based on the aircraft attitude data. It gives the elevation angle above or below the flight altitude. For better selection of the tie points, which is done manually, the contrast of the images is increased for better identification of recognizable structures of the cloud image. Fig. 6 illustrates the cloud geometry retrieval for a cloud scene from 19 September

2014. The selected cloud scene shows a strong convective cloud embedded in a stratiform cloud layer. After increasing the image contrast (Fig. 6a) several tie points with distinctive cloud features of individual clouds were selected. The same tie points are chosen in a second image taken about 10 seconds later. Choosing a short time interval helps to reduce the uncertainty of the method induced by cloud movement. From stereographic analysis of these tie points the distances to the cloud points (in km) are determined (Fig. 6b). From cloud distance and viewing elevation angle the height is calculated. Cloud top heights for this

case are in the range of 12 km, while the top of the stratiform layer is at about 6 km altitude. The corresponding isolines in Fig. 6c show quite a homogeneous horizontal distribution with negligible dependence on the azimuth angle for this particular cloud case. Therefore, the correlation between elevation angle and height is approximated by a polynomial fit of the third order as plotted in Fig. 6d. This fit is used to relate the elevation angles of the specMACS instrument to a cloud height. For all studied cloud cases of the flights AC10, AC13, and AC18, such simplified correlations between elevation angle and height are deter-

mined under the condition that the azimuthal dependence could be neglected which is fulfilled predominantly for sufficiently small cloud sections in the horizontal direction.

The accuracy of the cloud geometry retrieval depends on the distance to the observed cloud and the uncertainty of the angle determination. Uncertainties related to pixel selection are estimated with $\pm 5$ pixels (0.25°), which corresponds to an uncertainty of 130 m for a cloud distance of 30 km (maximum distance of observations). Additionally, the fitting method results in

mean deviations of 200 m. Overall, uncertainties between 200 and 300 m are calculated for the observing conditions during ACRIDICON-CHUVA.

## 4 Application

From the 14 scientific flights three days (AC10, AC13, and AC18) are selected with the best observation conditions for spec-MACS, namely: (i) no cloud layer above the observed cloud (no cirrus), which contaminates the spectral signature, (ii) high proportion of illuminated cloud parts in the vertical direction of the cloud, (iii) flight altitude that allows measurements of an

extended vertical region of the cloud considering the limited FOV of specMACS, and (iv) isolated clouds with recognizable structures for cloud geometry retrievals.

Phase profiles from AC13 representing polluted aerosol conditions will be compared to the two days with less aerosol pollution. Effects of aerosol conditions on the height and thickness of the mixed phase layer will be investigated. Second, it will be demonstrated how comparable the different observation strategies (cloud side, cloud top and in situ) are.

### 4.1 Case study for flight AC13 (polluted aerosol conditions)

During flight AC13 on 19 September 2014, several periods of cloud side observations are found. The flight track and the corresponding MODIS image are shown in Fig. 7. The 250 m resolution radiance of channel 1 (620-670 nm) of the MODIS overpass from 17:50 UTC illustrates the cloud coverage. The five colored lines denote the periods of cloud side observations between 17:50 and 19:00 UTC. The white arrows indicate the flight direction with specMACS pointing towards the clouds on the right hand side of the aircraft. The flight altitude for this one-hour flight track ranged between 5 and 10 km. As a result of cloud masking and cloud geometry analysis, the profile of the phase index for a cloud scene (section #A in Fig. 7) is shown in Fig. 8. The phase index is calculated in bins of 100 m in the vertical direction. The standard deviation is indicated by the error bars. A distinctive increase of the phase index is visible at 6.5 km altitude. Below that altitude a negative phase index indicating the liquid water phase is derived. Within the mixed phase layer the phase index increases sharply. The upper limit of the mixed phase layer is derived to be at 7.1 km. Above that altitude the variation of the phase index caused by changing particle sizes and viewing geometry is less pronounced.

Sixteen cloud cases are investigated for flight AC13. Each cloud scene is classified with respect to the phase state based on the profile of the phase index. Fig. 9a presents the statistics over all scenes. The background color of the scene number corresponds to the flight section as presented in Fig. 7. Obviously not all profiles show each of the phase states, mainly because of two reasons. First, the cloud particles may have the same phase state, or, second, the viewing geometry with respect to FOV, flight altitude, cloud height, and distance restricts the vertical range of the cloud observation. Overall, the depth ($\Delta z_{\mathrm{mix}}$) and vertical position ($z_{\mathrm{top}}$, $z_{\mathrm{bot}}$) of the mixed phase layer is highly variable for all cases: with $\Delta z_{\mathrm{mix}} = 1.2 \pm 0.4$ km (one-sigma standard deviation), $z_{\mathrm{bot}} = 6.2 \pm 0.3$ km, and $z_{\mathrm{top}} = 7.4 \pm 0.4$ km. Even for similar flight sections (as in #B and #D) the upper and lower limit of the mixed phase layer can vary by up to 900 m, which is larger than the uncertainty of the retrieval method. The corresponding temperature scale is displayed as non-linear secondary y-axis.

The variability of the mixed phase layer in depth and height within a single cloud cluster shows that the vertical distribution at least at the cloud edges is variable. In situ data are used to investigate if such a variability is also observed in the the more inner part of the cloud? In situ measurements of CAS-DPOL and hotwire data of the one-hour flight sequence (17:50-19:00 UTC) during AC13 are shown in Fig. 9b,c. The light dots are 1Hz data, while darker lines represent the 10th and 90th percentiles as well as the mean LWC and AF (squares), binned into 600 m altitude bins. Regions of mixed phase clouds are characterized by a decrease in LWC (decrease of the 90th percentile with altitude) and/or an increase in AF. In these in situ measurements of LWC and AF, the mixed phase region extends between 6.4 and 8.7 km. However, the profiles shown in Fig. 9b,c are based on data sampled over the entire cloud cluster including clouds at different stages of evolution, and profiles of individual clouds cannot

be derived from this data set, which prevents a direct comparison of the in situ and remote measurements. The asphericity of cloud particles in the size range 20 - 50 $\mu$m derived from NIXE-CAPS is shown in Fig. 9d for the one hour time frame of

the cloud observations. The data are classified as listed in Tab. 2. The heterogeneity of cloud particle asphericity between 5 and 8 km altitude is observed from its variable classification during the ascent around 18 UTC with solely spherical particles (could be also related to small spherical ice particles) and during the descent between 18.25 and 18.80 UTC with spherical and aspherical particles. Mainly aspherical particles of Group II are observed, indicating the existence of large ice particles with sizes larger than 50 $\mu$m. Except for two single cases, a larger number of spherical particles (open green circles) is observed up

to an altitude of 8 km. From the descent flight track the position of the mixed phase layer is estimated between 6 and 8 km. For example, a closer look at the asphericity is taken for the time range between 18.28 and 18.34 UTC (Fig. 9e). At a constant flight level near the upper boundary of the mixed phase layer the occurrence of spherical and aspherical particles is somewhat separated. While mainly spherical particles are observed during this selected flight section for vertical wind speeds between $\pm1$ m s$^{-1}$, there are also segments with higher vertical wind speeds (between -3 and 5 m s$^{-1}$). For this section (around 18.315

UTC) large aspherical particles representing ice particles were also measured. This suggests that the vertical distribution of ice and liquid particles is affected by up- and downdrafts within a convective cloud, and therefore it is not homogenous inside the same cloud.

After showing these results from in situ and cloud side measurements, we also present retrievals of the phase state based on cloud top MODIS observations. In Fig. 9f the frequency of liquid and ice phase observations for altitude bins of about 200 m

is presented. Fully developed deep convective clouds with cloud tops between 10 and 14 km (classified as ice cloud) and low level cumulus clouds up to 6 km (liquid water clouds) are detected. Cloud phase information from the assumed phase transition layers is not available in Collection 6. Nevertheless, there are some levels with low frequency classified as ice and liquid phase (8 - 11 km), corresponding to temperatures between -20 and -42°C. In particular, at very low temperatures (lower than -38 °C) the presence of liquid particles can be excluded even for situations of homogeneous freezing. In fact small ice particles may be

misinterpreted as liquid particles by the retrieval algorithm at this level (Järvinen et al., 2016).

We applied the ensemble method to derive profiles of the effective particle size and to estimate the glaciation height and temperature following the retrieval technique of Yuan et al. (2010) for the MODIS scene. For better comparison, the brightness temperature as vertical coordinate is converted to altitude. Cloud top brightness temperatures (at 11 $\mu$m, corresponding to MODIS Channel 31) are simulated for variable cloud top heights and an atmospheric profile of temperature and humidity as

measured by the aircraft. The best agreement of simulated and measured cloud top brightness temperature is used as proxy of the cloud top altitude. The result is presented in Fig. 9g. The particle size is increasing with altitude up to a height of about 9.0 km (horizontal black line). This level is assumed as glaciation height, the upper level of mixed phase layer. The standard deviation of the binned (2 K bins in brightness temperature) particle sizes (horizontal error bars) is significantly larger for altitudes below 11 km, indicating a larger variability of the cloud particle size and a smaller statistics. Furthermore, a second

but smaller peak of the particle size is found at about 6 km altitude. From the conceptual model of cloud particle size profiles inside a DCC (e.g., Rosenfeld and Woodley (2003)) it might indicate the bottom of the mixed phase layer, when cloud particle size starts to increase. However, this increase is less pronounced than presented in Rosenfeld and Woodley (2003).

Comparing the glaciation height from MODIS with NIXE-CAPS in situ data and results from specMACS observations shows a deviation of about 1.0 - 1.5 km between the different retrieval techniques and observation strategies. However, the mean profile over the entire cloud cluster derived from CAS-DPOL measurements exhibited a similar glaciation height (of about 8.7 km) as found from the MODIS data. This shows that the satellite-based ensemble method may be representative for a large cloud field. But for individual clouds NIXE-CAPS and specMACS measurements have shown lower glaciation heights. The most likely reason is related to the fact that the ensemble method relies on cloud top observations of growing clouds in different stages of evolution. As shown in Fig. 9g mainly particle sizes between 22 and 27 $\mu m$ are derived indicating that the profile is dominated by measurements of clouds in the mature stage. At this stage, the particle phase may be altered by up- and downdrafts within the clouds as was shown in Fig. 9e. This leads to an enhanced horizontal variability of the cloud phase state which cannot be resolved by passive remote sensing from cloud top observations. Another, but minor reason of the discrepancy between ensemble method and NIXE-CAPS / specMACS measurements is related to the retrieval uncertainty of the effective cloud particle radius. While scattering properties are well defined for liquid water particles, they are variable for ice particles due to differing habits and crystal shapes (Eichler et al., 2009). This gets even more complicated for cloud tops where phase transition starts. Additional retrieval uncertainties of the particle size directly contribute to the derived profile of $r_{eff}$.

## 4.2 Comparison with less polluted conditions

Profiles of the phase state for two other flights (AC10 and AC18) performed under moderate aerosol conditions are presented in Fig.10. On both days the number of complete profiles showing liquid, ice and the mixed phase layer is smaller compared to AC13. Mainly low level clouds or cloud parts with liquid water were observed during AC18. The lower boundary of the mixed phase layer is estimated to be about 5.5 km (-4°C). From NIXE-CAPS measurements, large aspherical ice particles are found down to 5 km (-1°C), whereas spherical particles assumed to be as liquid water were observed up to 8.7 km. In contrast, the specMACS data exhibit ice phase down to 7.7 km. As in the case of AC13, the cloud top MODIS retrievals of the phase state only distinguishes between liquid and ice phase. Because of the low statistical significance of clouds with cloud tops higher than 6 km in the MODIS data, no profile of effective drop radius is derived.

On flight AC10 no in situ data within mixed phase clouds were obtained. The MODIS phase product shows ice cloud tops between 11 and 15 km altitude and liquid water clouds up to 4.5 km. But the profile of the effective particle radius based on the ensemble method retrieval gives a glaciation temperature of 260 K, which corresponds to an altitude of about 7.2 km. The specMACS profiles as plotted in Fig. 10b show highly variable mixed phase layers. While clouds #1 - #3 with cloud tops between 6.0 - 6.8 km are classified as liquid water clouds, the profiles of the phase index of clouds #4 - #6 reveal also the existence of ice particles between 4 and 7 km altitude. As illustrated in the RGB image taken by the GoPro camera (Fig. 10c), cloud #3 and #4 are in close vicinity but in different states of evolution. The diffuse cloud areas with smoother texture in the GoPro image of cloud #4 indicate precipitation, which explains positive phase indices down to 4 km corresponding to more than 0°C. As Fig. 10d shows, the phase index can vary significantly for one altitude level depending on the occurrence of precipitation. Consequently, the individual state of evolution of each cloud determines the distribution of particle sizes and phase state. Also local strong downdrafts can transport ice particles into lower levels, which will be interpreted as mixed phase

layer from the cloud side observation perspective. Due to the horizontal variability of cloud phase inside a cloud cluster for example caused by up- and downdrafts, in situ measurements may only reveal liquid phase particles. A direct comparison between the observation strategies is subject to restrictions because of temporal and spatial variability of cloud properties in convective systems.

From theory, the mixed phase layer is expected to be higher for polluted aerosol conditions than for cleaner aerosol conditions,
which can partly be confirmed by comparison of the three cases. We find from cloud side observations, that the lower boundary altitude of the mixed phase layer tends to be higher for polluted conditions (AC13: 6.0 - 6.5 km) than for the moderate case of AC18 (5.6± 0.2 km), while the upper boundary is shifted from 6.8± 0.2 km (moderate case AC10) to 7.4±0.4 km (polluted case AC13).

## 5   Conclusions

The vertical evolution of deep convective clouds is linked with the phase transition from liquid water via the mixed phase to ice. Aerosol particles may alter the radiative effects of cloud particles (also with respect to their phase state), their lifetime and the formation of precipitation. This study documented the vertical distribution of the cloud phase for different aerosol conditions as measured during the ACRIDICON-CHUVA campaign over the Brazilian rainforest in September 2014. Our approach applies a retrieval method to quantify the height range of the mixed phase layer. Cloud side observations performed
by an imaging spectroradiometer were used to determine a phase index based on differential absorption by ice and liquid water in the spectral range between 1550 and 1700 nm. Negative values of the phase index indicate liquid particles, whereas ice particles are characterized by a positive phase index. It was shown by 3D radiative transfer simulations that the mixed phase zone is characterized by a significant gradient in the profile of the phase index. A cloud mask method to discriminate between shadowed and illuminated cloud regions was presented to exclude the shadowed areas in the cloud scene. 3D radiative transfer
simulations were performed to validate the approach. Since the imaging spectroradiometer delivers spectral radiation data as a function of viewing zenith angle, the derived mean vertical profiles of the phase index needed to be referenced to altitude ranges. For this purpose, stereographic methods were applied to collocated GoPro camera observations to estimate the cloud geometry in terms of cloud height profiles and distance to the aircraft.

The profiles of several individual clouds were classified with respect to their zones of phase states. Depending on the viewing
geometry and cloud distance, layers of pure liquid and ice phase, as well as phase transition layers were identified. It was found that the height and thickness of the layers of phase transition were variable (900 m in upper and lower limit) even for one compact cloud cluster measured during flight AC13 with polluted aerosol conditions. Here first ice particles were found at temperatures between -3 and -9 °C, while full glaciation was observed between -10 and -20 °C. For moderate aerosol conditions, only few cases exhibited liquid water, mixed phase, and ice phase, which limited the statistical significance of the
comparison with AC13. However, comparing the glaciation heights of AC10 (6.8± 0.2 km ) and AC13 (7.4±0.4 km) we found an indication of an increase of glaciation height and a decrease of glaciation temperature for polluted aerosol conditions. With respect to the occurrence of first ice particles, the lower boundary of the mixed phase layer was derived with 6.0 - 6.5 km for polluted conditions, whereas for AC18 the altitude was shifted down to 5.5 - 6.0 km, which agrees with theory.

Also, in situ measurements of the cloud particle size distribution together with the asphericity of particles between 20 and 50

$\mu$m, measured by the cloud spectrometer NIXE-CAPS, were used to estimate the cloud's phase (Costa et al., 2017). Aspherical particles can be considered as ice, whereas spherical shapes are related to liquid droplets or spherical ice. In contrast to cloud-side remote sensing, in situ observations represent point measurements within the cloud. Therefore, in situ profile information of an individual cloud is a combination of data from different states of evolution. Consistent results of mixed phase zone levels were found from specMACS and NIXE-CAPS measurements, for the flight AC13 with most individual cloud cases showing pure liquid, mixed phase layer and pure ice phase.

Additionally to in situ and cloud side measurements, the glaciation temperature was derived applying an ensemble method based on MODIS data, which assumes time–space–exchangeability for a cluster of clouds with different states of evolution. For the polluted and moderate flights, retrieval results of the effective particle size at cloud top were combined into one single profile. For flight AC13 the glaciation height of 9.0 km (-26°C), defined by the level of maximum particle size, deviates from the in situ (8 km) and specMACS results (6.8 - 8.2 km). However, for the moderate aerosol case the glaciation height was much lower at about 7.2 km (-13°C), similar to the height derived from specMACS observations (7 km).

The presented study has shown that the occurrence of ice particles and the level of the mixed phase layer vary by several hundred of meters even for similar atmospheric conditions. Two cloud cases in close vicinity clearly show different cloud phases at the same altitude. It is assumed that downdrafts and falling precipitation in well-developed clouds alter the retrieval results of the phases' vertical distribution. It is concluded that the assumed time–space–exchangeability used in the ensemble method can give a simplified picture of the vertical distribution of the phase within a field of convective clouds of different stages of evolution. Particularly, cloud tops where phase transition (from liquid to ice) starts and ends needs to be observed by the satellite to profile the thermodynamic phase. The number of these observations has to be significant, since the particle sizes are averaged over a larger domain. So, in general the ensemble method can give an indication when phase transition arises for the first time. However, for estimation of the cloud phase profile at a later stage of the DCC evolution, in situ and also cloud side remote sensing might be the better observation strategy, when phase distribution is altered for example by up- and downdrafts.

Planned future studies include observations of individual convective clouds to document their evolution from growing to mature and finally to dissipating stages of development. We intend to deploy our sensor on ATTO (Andreae et al., 2015), which is 325 m high and is used to perform continuous monitoring of chemical, meteorological and aerosol parameters. The ATTO tower is located near the Equator (a region with daily occurrence of DCCs in a highly variable environment with respect to concentrations and types of aerosol particles) and will serve as an ideal platform for upcoming studies.

*Acknowledgements.* The ACRIDICON-CHUVA campaign was supported by the Max Planck Society (MPG), the German Science Foundation (DFG Priority Program SPP 1294), the German Aerospace Center (DLR), the FAPESP (Sao Paulo Research Foundation) grants 2009/15235-8 and 2013/05014-0), and a wide range of other institutional partners. It was carried out in collaboration with the USA–Brazilian atmosphere research project GoAmazon2014/5, including numerous institutional partners. We would like to thank Instituto Nacional de Pesquisas da Amazonia (INPA) for the local logistic help prior, during, and after the campaign. Thanks also to the Brazilian Space Agency (AEB: Agencia Espacial Brasileira) responsible for the program of cooperation (CNPq license 00254/2013-9 of the Brazilian National Council for Scientific and Technological Development). The entire ACRIDICON-CHUVA project team is gratefully acknowledged for collaboration and support. Evelyn Jäkel gratefully acknowledges funding of parts of this work by the German Research Foundation (DFG) under grant number (JA2023/2-2).

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

**Table 1.** Summary of presented flights with cloud side observations during the ACRIDICON–CHUVA campaign. The ranges of flight altitude and time refer to the studied cloud cases.

| Flight number | AC10 | AC13 | AC18 |
|---|---|---|---|
| Aerosol conditions | moderate | polluted | moderate |
| AOD (MODIS) | 0.4 - 0.5 | 0.5 - 0.6 | 0.3 - 0.4 |
| Number of cloud cases | 9 | 16 | 10 |
| Flight altitude range (km) | 7.4 - 10.4 | 5.2 - 9.3 | 1.4 - 14.0 |
| Time range (UTC) | 17:25 - 19:20 | 17:55 - 19:00 | 15:30 - 20:30 |

**Table 2.** Cloud flag description of the NIXE-CAPS asphericity product after Costa et al. (2017). Group I: total concentration of particles 3-50 $\mu$m is larger than 3 cm$^{-3}$, Group II: total concentration of particles 3-50 $\mu$m is smaller than 1 cm$^{-3}$ and total concentration of particles with size larger than 50 $\mu$m is larger than 0 cm$^{-3}$.

| Cloud flag | Temperature range ($^\circ$C) | Description |
|---|:---:|---:|
| 1.0 | $> 0$ | no aspherical particles detected; liquid |
| 1.1 | $> 0$ | aspherical particles detected - could be ice or ash particles |
| 2.0 | $0 > T > $ -38 | no aspherical particles detected; liquid |
| 2.1 | $0 > T > $ -38 | aspherical particles detected, group I; mixed phase |
| 2.2 | $0 > T > $ -38 | aspherical particles detected, group II; ice |
| 3.0 | $< -38$ | below homogeneous freezing threshold: all ice, no asphericity criterion; ice |

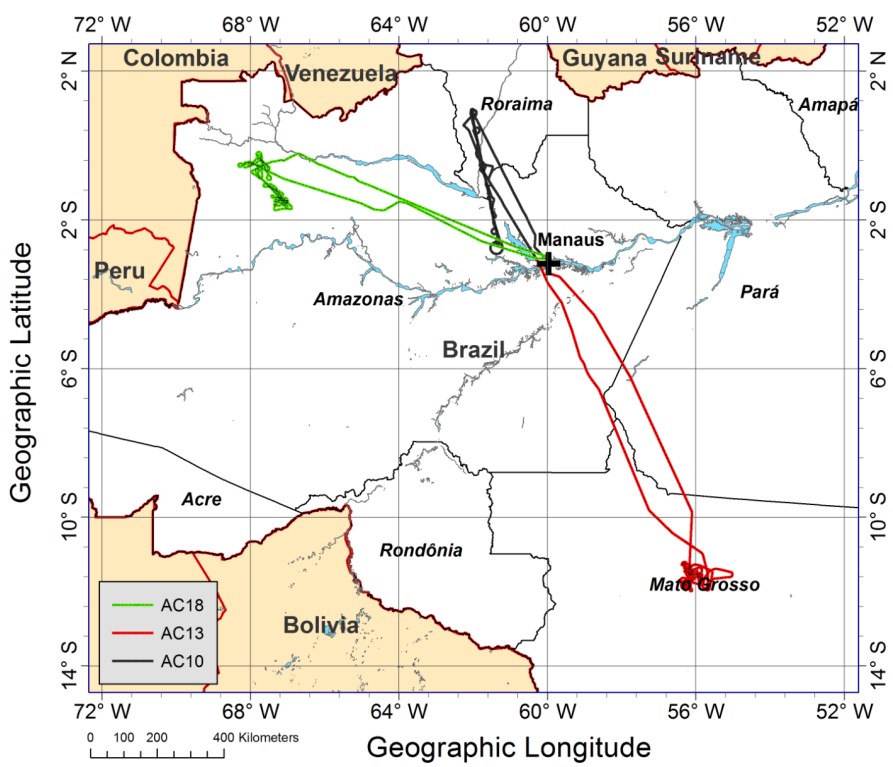

**Figure 1.** Flight tracks of AC10 (black), AC13 (red), and AC18 (green). The city of Manaus is indicated by the black cross.

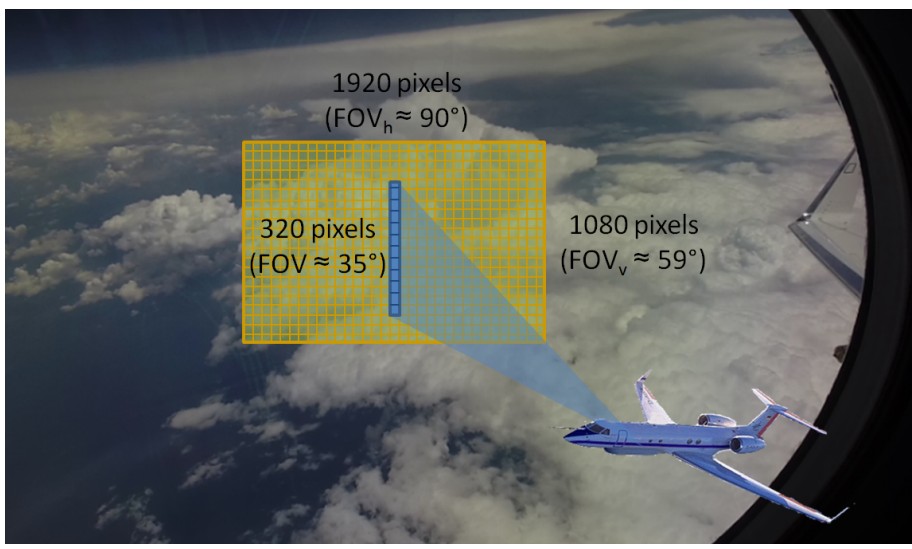

**Figure 2.** Schematics of cloud side observations by the imaging spectrometer specMACS (SWIR camera) and the GoPro camera. The individual field of views (FOVs) and corresponding number of spatial pixels are illustrated.

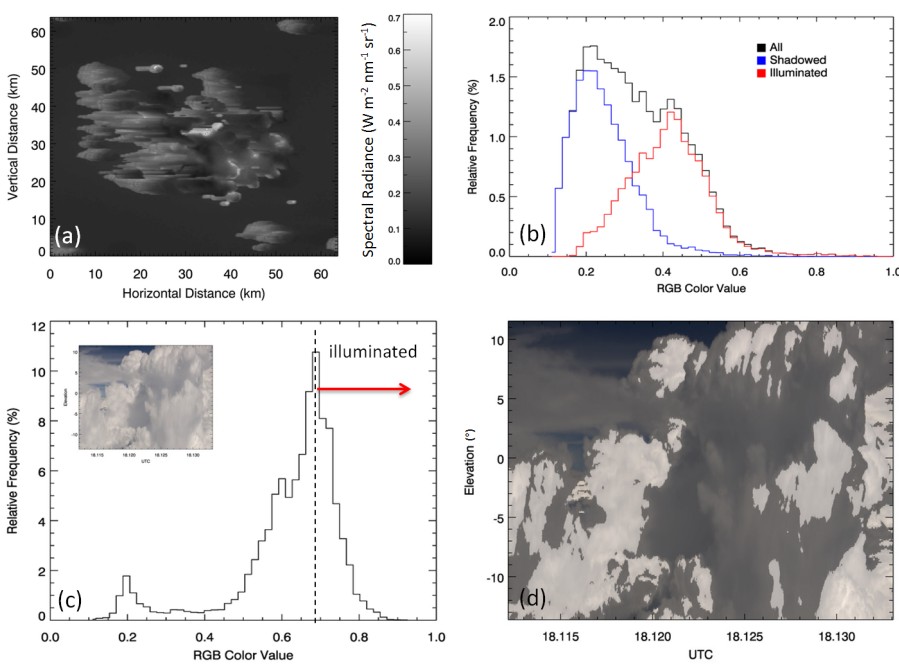

**Figure 3.** (a) Field of RGB color values from simulated spectral radiances for cloud side viewing geometry with a sensor elevation angle of $10°$ and a relative azimuth angle of $60°$. (b) Histograms of RGB color values of the field shown in (a). (c) Histogram of RGB color values for a measured cloud scene shown in the inset. (d) Identified illuminated cloud sides of the observed cloud scene are highlighted in brighter colors.

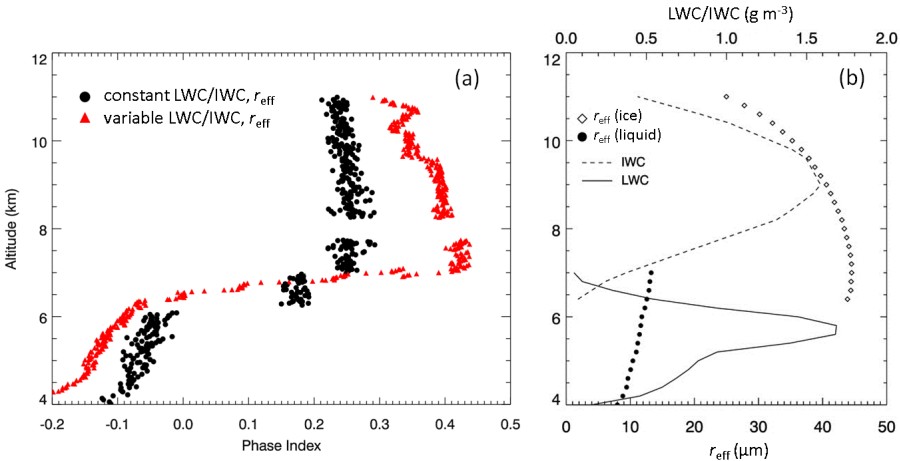

**Figure 4.** (a) Phase index derived for simulated clouds with variable LWC/IWC and effective radius and fixed values of microphysical properties. (b) Profile of corresponding cloud with variable LWC/IWC and $r_{\mathrm{eff}}$.

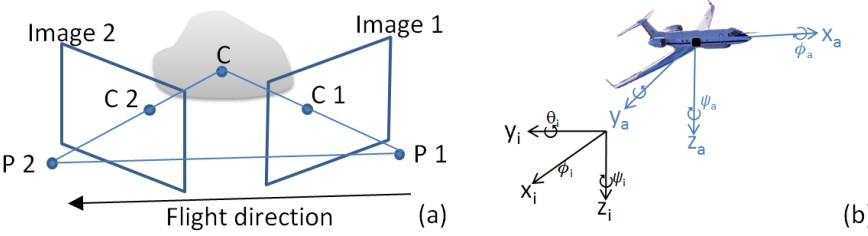

**Figure 5.** (a) Schematics of stereo-photogrammetric observations of cloud point C from aircraft position P1 and P2 with projected image points C1 and C2. (b) Illustration of aircraft and camera coordinate systems.

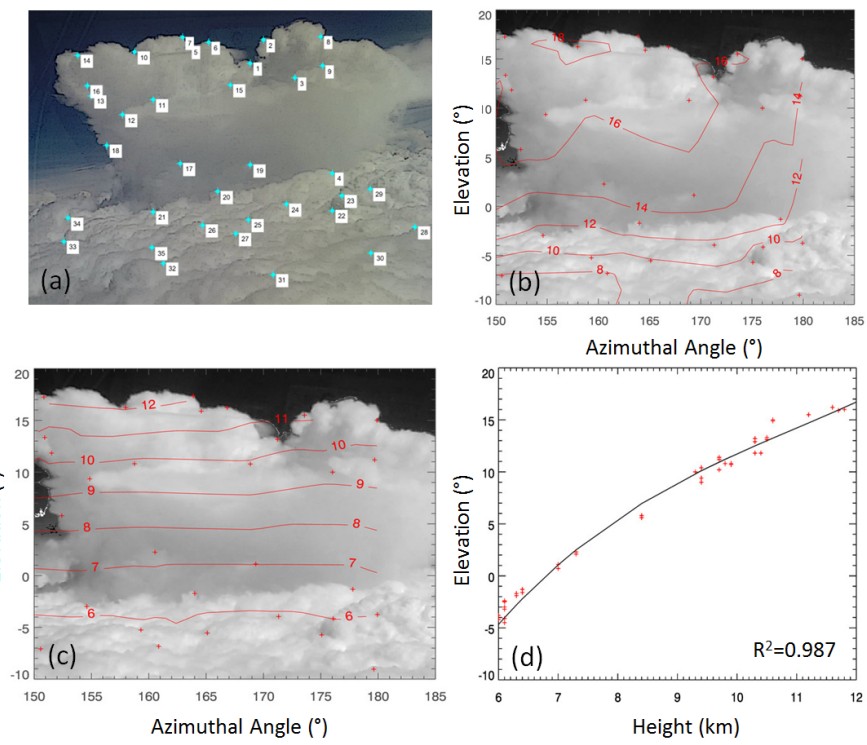

**Figure 6.** (a) Cloud image from GoPro camera with enhanced edges and selected tie points from 19 September 2014. (b) Calculated distances in km to the individual cloud points for the cloud scene displayed as isolines. (c) Corresponding isolines of calculated heights. (d) Relationship of height and elevation angle derived for the cloud case including a polynomial fit with a correlation coefficient of $R^2 = 0.987$.

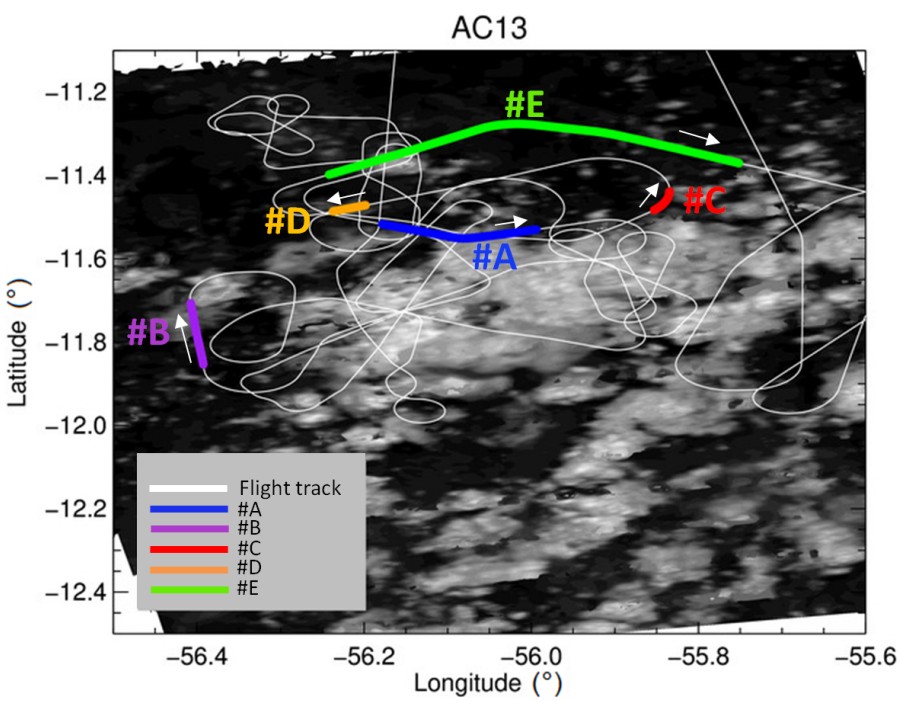

**Figure 7.** Flight track (white line) and selected time periods of cloud side observations during AC13 (19 September 2014). Additionally, the 250 m resolution product for channel 1 (620 - 670 nm) of the Aqua-MODIS instrument from 17:50 UTC is shown in the background. Figure is similar as presented in Jäkel et al. (2016).

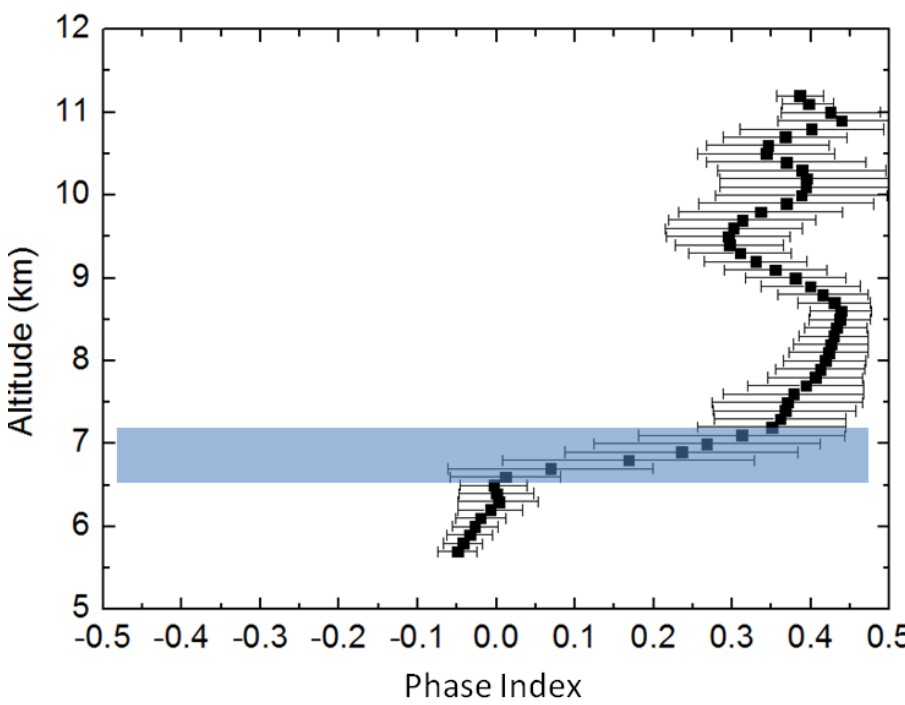

**Figure 8.** Mean phase index profile for cloud scene shown in Fig. 6. The mixed phase layer is indicated by the colored area.

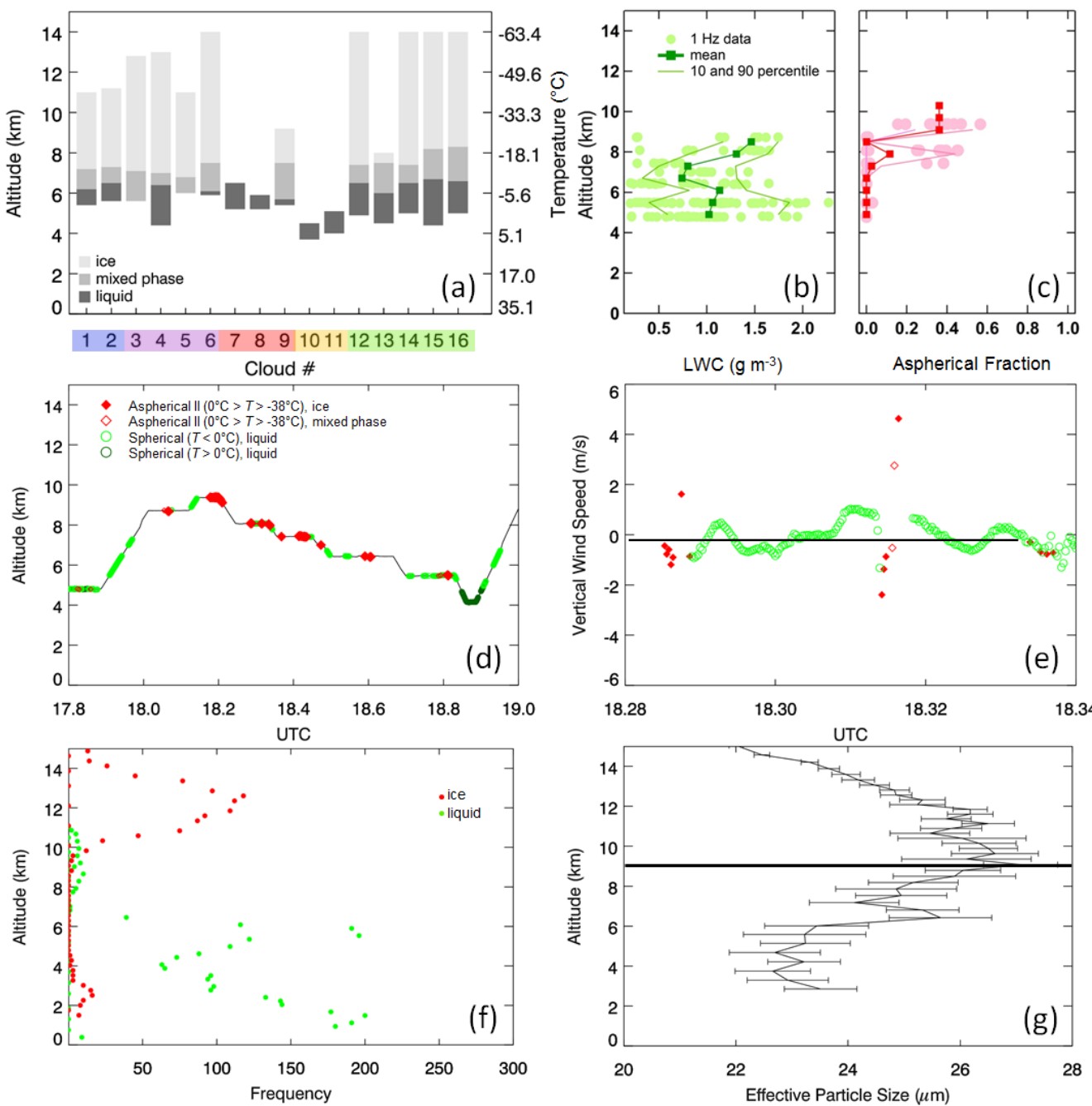

**Figure 9.** (a) Phase classification of studied clouds based on specMACS observations during flight AC13. (b) Profile of LWC measured with the hotwire probe between 17:50 and 19:00 UTC. (c) Aspherical fraction derived from CAS-DPOL in situ data. (d) NIXE-CAPS in situ measurements of liquid, mixed phase and ice, see Table 2 for definitions. Note, that time is given in decimal hours. (e) Short horizontal flight section in the upper part of the mixed phase layer showing the relation of vertical wind speed and classified asphericity of cloud particles. Symbols as in (d). (f) Classification of cloud phase (ice or liquid) from MODIS observations of cloud tops. (g) Mean profile of effective particle radius from ensemble method based on MODIS retrieval data of cloud top effective radius. The black horizontal line indicates the level of largest ice particles.

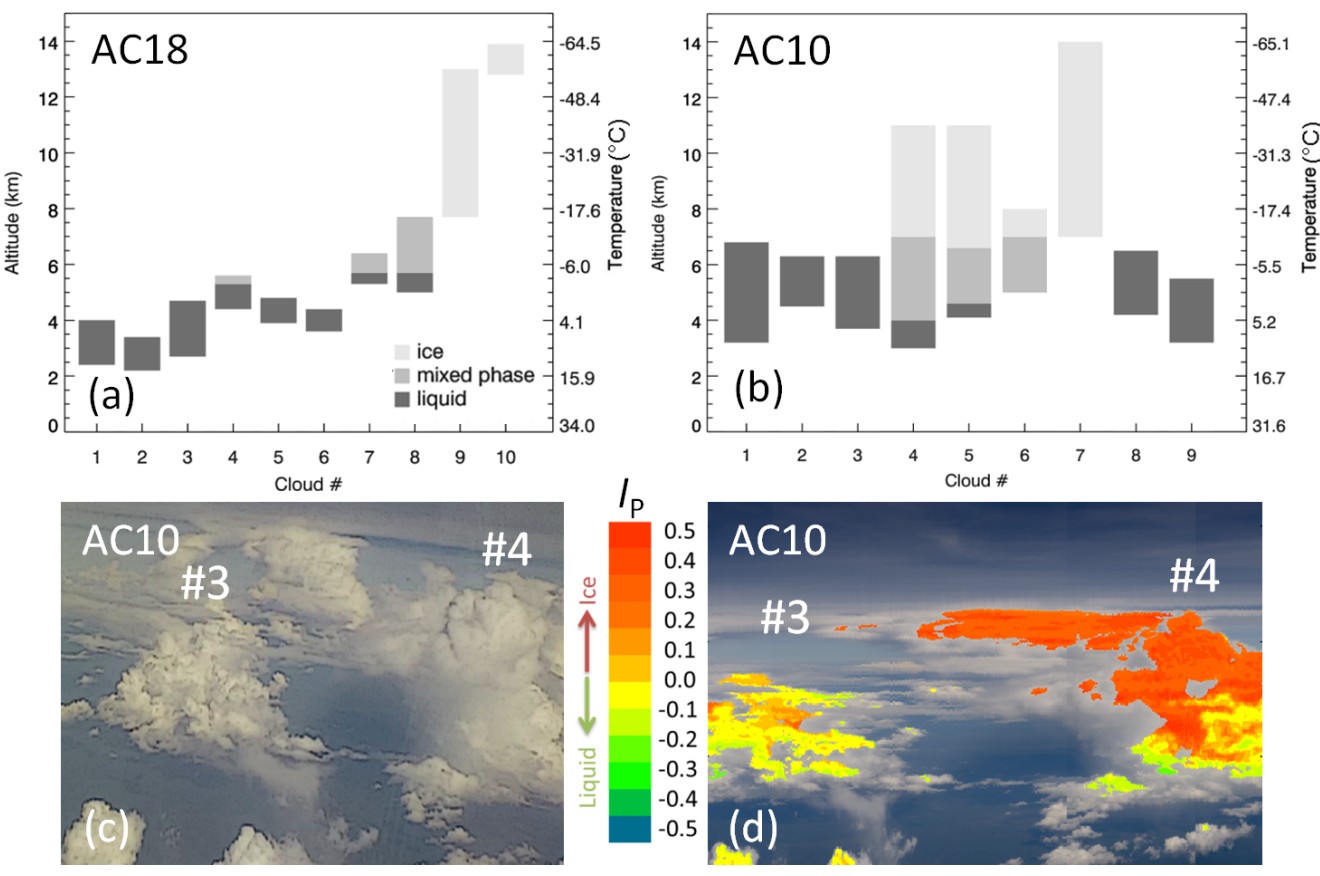

**Figure 10.** (a) Phase classification of studied clouds based on specMACS observations during flight AC18. (b) Same as (a) but for AC10. (c) GoPro image of cloud scene during AC10. (d) Phase index as derived from specMACS during AC10 for illustrated clouds from (c).