# Peer review of "Vertical distribution of the particle phase in tropical deep-convective clouds as derived from cloud-side reflected solar radiation measurements"

_Atmospheric Chemistry and Physics, 2017_

## Referee Comment (RC1) · Anonymous Referee #1 · 6 Mar 2017

General comments: The analysis and modeling of deep convective clouds (DCC) and their vertical distribution of certain parameters is an ongoing research topic. The manuscript describes many methods and aspects of the possible parameters derived by optical measurements in a special configuration measured sideways from aircraft. The phase state of the cloud particles are derived by a method already published by the author. Nothing new about that. The only new part is the geometric retrieval.While this configuration is good for this kind of case studies it is not a standard configuration compared to satellite instruments or other airborne applications. Hence this case study can give precious information if thoroughly compared and validated to standard

retrievals and model comparisons.

Questions: - The combination of geometric retrievals and hyperspectral measurements is new and can give additional cloud structure information. In addition its an important information to analyze angle dependent reflectance properties of the DCC relative to the Sun, but the derived parameter of cloud distance from the geometric retrieval is only one single parameter. The observed area by the imaging instrument depends on the field of view, cloud structure and distance and could lead to a 3D cloud structure, but the simple assumption of a homogenous vertical cloud area within the field of view of a single imaging pixel might lead to errors in the analysis of cloud scattering effects.

- A cloud masking procedure is introduced to distinguish between directly reflected areas of the cloud and diffuse shadow regions. The analysis of the manuscript is restricted to directly reflected areas only, which in fact is a sum of direct and diffuse light.

- Why does the described method of the distribution of the cloud phase does work only for direct reflected light of the Sun?

- What is the influence of the diffuse light?

- It would be nice to see some more direct and detailed comparisons to the methods of Marshak (2006, reference missing) or Zinner (2008) , MODIS, possibly Cloudsat and insitu. The description of Figure 9 could be in much more detail and as the major part from my point of view this is worth more than just one page.

Detailed comments: Page 2 line 5: A mixed phase state of water does not exist. I would rather describe it as an area of phase transition levels from existing state phases eg. from liquid to ice which can vary in temperature gradient, altitude and vertical depth (line 17 is here more precise than 5)

Page 2 line 13: . . . (more aerosol particles . . .)

Page 3 line 17: Why did Martins and Marshak use a different Wavelength? With Spec-

MACS it could be used as well and compared.

Page 4 line 14: Temperature profiles are mentioned and it would be very helpful to have some graphs. Sidewards looking IR-camera would be interesting.

Page 5 line 2: Again a comparison to the method of Marshak, . . . is possible with SpecMACS.

Page 5 line 29: mixed phase layer . . . → phase transition layer

Page 5 line 29: A retrieval of cloud particle size of the measurements would demonstrate this sentence Page 6 line 11: Please explain this statement, if its true. Please compare state of polarization with Mi-Theory.

Page 6 line 23 and 30: detection limit unclear. >1 cm-3 or 0.3 g/m-3

Page 7 line 3: Please explain the adjustment of the temperature, humidity, . . . profiles

Page 7 line 11: adjustment of the aerosol profile?

Page 7 line 19: As mentioned before. Why is the diffuse light restricted to shadow regions or does it have the same amount in the other regions as well?

Page 7 line 21: Here we have some weak indications why the diffuse light in shadow regions are not used in this study. While a thoroughly radiative transfer simulation can include the influence of ground reflectance, surface albedo in this manuscript cant be taken into account because of this influence. Why is that and a view sentences later the influence of the surface cant be seen in the airborne data? The reasoning in this part of the manuscript is somehow very weak.

Page 8 line 29: Where does this formula and the constants come from? I would propose to use a spectrum to rgb conversion via a CIE 1931 color space. SpecMACS has a broad spectral range and a large number of spectral channels why not using them? This would reduce noise as well.

Page 8 line 1, ...: How is the histogram of the RGB values converted or evaluated to the frequency distribution? Please explain in more detail. Where does the relative and absolute frequency come from in fig. 3? Why are the simulated once in Fig 3b absolute and the measured once in Fig 3c relative? The calculation of a single RGB value with the formula is used to find the threshold of "directly !" illuminated pixels. What are the model simulations for if you dont use them?

Page 8 line 3: What is the max height of the model domain?

Page 8 line 16: What is a relative azimuth angle of exactly 68 degree with a changing attitude and Sun elevation during airborne missions?

Page 8 line 22,23: The simulation shows an increase in cloud particle size in Fig 4 for that region. What is wrong?

Page 8 line 28: How does this simple formula compare to the methods from Marshak, Martins and Zinner?

Page 9 line 5: Is the combined Ip profile a simulated or measured profile. I dont understand how the combined profile is calculated and where it comes from.

Page 9 line 7: three phases ?

Page 9 line 10: What is a pronounced absorption, of what?

Page 9 line 12: Each cloud height → The cloud vertical structure is ...

Page 9 line 14: To derive the particle size is first mentioned here. Is that the goal or what is the reason? A look up table would do as well, please look at AMT Zinner 2016.

Page 9 line 15: What is a more realistic cloud? Are the other clouds not realistic?

Page 9 line 17: What is the first case?

Page 9 line 18: The transition layer is characterized by a strong increase in particle size and change in the value of phase index. See Fig 4b (simulations) and Fig. 8

(measurements).

Page 9 line 24: I assume that we have a polluted and a clear case, but its not clear in this part of the manuscript. Here we have only two cloud cases, one with fixed microphysics and one with changing cloud properties. Please clearify.

Page 10: Geometry is Ok, but could be shorter. Except a real 3D cloud structure would be the final product.

Page 12 line 12: A profile and comparison of remote sensing and insitu droplet size would be interesting. A sharp transient of the droplet size shows the transition layer.

Page 12 line 30: mixed phase levels → phase transition levels or better layer

Page 13 line 20: Why are liquid water data from up to 8.7 km not shown

Page 14 line 2: A temp profile is missing.

Page 14 line 25: three phases ?

Page 14 line 29: Is there only one polluted case during the whole campaign?

Page 14 line 30 bottom: Low statistics? Are those 2 flights analysed in this study the only possible ones of the whole campaign?

---

## Referee Comment (RC2) · Anonymous Referee #2 · 10 Mar 2017

REVIEW of "Vertical Distribution of the phase state of particles in tropical deep-convective clouds as derived from cloud-side reflected solar radiation measurements"

General comments: In this paper, the profile of thermodynamic phase of tropical deep convective clouds is derived from passive shortwave reflectometry with a lateral viewing geometry. Whereas the phase discrimination is directly determined from spectral imagery by way of a previously established near-infrared bi-spectral retrieval method, the altitude registration is established through an additional stereo inversion based on a standard RGB camera. The altitude-registered profiles of individual clouds are put

into the context of in-situ measurements as well as satellite observations.

The main intent and take-home message remain a bit unclear. The introduction starts off by summarizing aerosol effects specifically on deep convection, emphasizing the location and width of the mixed-phase zone as an indicator for the potential influence of aerosols on deep-convective cloud development.

One of the lingering questions with respect to the observational dimension of this problem is whether time and space can indeed be regarded as interchangeable to study the problem with the so-called ensemble approach where single-image satellite snapshots of multiple clouds at different development stages are statistically linked to their temporal development. Detailed airborne observations could provide further evidence that this approach does indeed work, and thus justify the use of satellite observations in future studies. But it is unclear whether this study intends such work. Instead, it seems to seek consistency between retrievals from airborne imagery, in-situ data, and concomitant satellite images, without really drawing conclusions with regard to aerosol influences on vertical development due to the limited number of samples.

It would help the paper if the objectives were laid out a bit more clearly, perhaps through a number of questions that are answered in the conclusions. To a certain degree, the reader is left wondering about the distinct advantages of aircraft over satellite observations (aside from resolution). Why is a lateral viewing geometry superior to satellite imaging (primarily from above), and what is the advantage of resolving even the smallest convection events? What parameters can we get from aircraft observations that satellites do *not* provide – and that are yet critical to advance theory? Some of these questions may be answered implicitly, but if relevant, they should be addressed in the paper, preferably with support from the measurements.

I believe that minor modifications can go a long way towards addressing some of my general comments above. Below, I offer specific comments in sequential order, which are intended to improve the next version. I would be willing and happy to look at the

revised manuscript.

Sequential comments:

p3,L5: I don't see the relevance of the cited paper (Cahalan, 1994) and the associated science (plane-parallel retrieval assumptions) in this context.

p3,L17+L30: These are places where the manuscript could outline how the specific work fits into the larger context that was set up previously. Currently, this page in particular looks like a list of work done by the authors and predecessors (e.g., Martins Rosenfeld), with only tangential connection to the motivation from the previous page(s). This is not a deal breaker for the manuscript, but it would be better to see how the listed work serves a number of outstanding questions related to the introductory comments earlier on.

p4,L4: In the description of the manuscript's structure, it seems that the goal is to compare remote sensing derived cloud profiles to MODIS and in-situ data, which would make the manuscript more appropriate for AMT than for ACP. If, however, there are some higher-level goals that address some of the questions brought up above, this should be made clearer.

p4,section 2.1: A table with flights and clouds cases would help.

p5,L12-18: The manuscript should elaborate on the stereo algorithm a little more. Also, L21, should this be "assign" instead of "allocate"?

p6,L23: The description of the aspherical fraction is a bit unclear; what is measured, and what is derived?

p6,L30: Why is the threshold this high? 0.3 g/m3 seems excessive, considering that a typical BL cloud top-level LWC is 1 g/m3.

p7,l1: Why is it necessary to perform 3D calculations? Only because of the geometry, or because deviations from standard 1D models are expected? If so, what are they

(aside from shadows)?

p7,l6: Rayleigh scattering —> molecular scattering?

p9: Here it becomes quite difficult to understand what the authors are after - the position/width of the mixed layer? Is that the purpose of the simulations? Are they done to only replicate the earlier study, or are they something new?

p10: The geometry retrieval description is rather cryptic. Would it help to cite work related to MISR, or is the method unrelated? What happens if the cloud moves during two consecutive images (used for the stereo method)? Also, elaborate on p10,l18-19, and state with respect to which coordinate system the "elevation angle" is provided.

p10: Others circumvented the whole (rather difficult) stereo algorithm by including an IR channel. Why was that not done here? Were these measurements simply not available? And why did the authors prefer the more complicated method to the simple IR imager?

p11, Applications section. What is it that the paper seeks to find out? Refer to the main question here, at least at the beginning? This whole section reads a little bit like a listing of results with no specific purpose. Quite surely there is one, and that should be clarified more.

p12: It is a bit unclear what the in-situ measurements really have to offer here if "the direct comparison of in situ and remote measurements is difficult". Really, the in-situ measurements should serve as validation for the remote sensing, but what do we learn if that doesn't work? Is it still worth using the in-situ data? If the comparison does not work out, what does that mean for the initial hypotheses (if there is one: perhaps a question about the interchangeability of pixel/time mentioned earlier?)

p13: The in-situ data puts ice higher than remote sensing. Which is right? What do we learn here about the representativeness of satellite data and *its* consistency with the aircraft measurements?

p13,L34 (top): So what phase *does* MODIS get for 6km? The shown results are certainly not liquid drops, given the large size.

p13,L12-14: Here we get some potentially important conclusions, which should be expanded an elaborated on. What is the significance of this finding? What can the satellite-based ensemble method do, and what not? Do in-situ and remote sensing observations from the aircraft tell two different stories?

p14,l5: Does this explain the discrepancy between in-situ obs and remote sensing?

p14,l16/17: distinctive *change* in gradient, or simply "significant gradient" (change of gradient is a gradient of a gradient...)

p14,L21: Here the question is again why this was chosen over IR imaging.

p14,L31: Earlier, the authors said that the aircraft measurements are not statistically significant to prove/disprove theory. This statement here is not meant to be the main finding of the manuscript, is it? Wouldn't satellite data be more suitable to put this on a statistical basis? If so, what would then be the purpose of the aircraft measurements? This may be obvious, but it would help the reader to understand this point.

p15,L38(top): The results from remote sensing and in-situ are not really consistent (as discussed earlier by the authors, and noted by the reviewer).

p15,l3: "invariance of space and time" does not seem an appropriate way to describe the assumptions of the ensemble method. It's really spatial statistics vs. temporal evolution, isn't it? Secondly, the manuscript now divulges that it did seek to study the aerosol effect on deep convection - or is this a statement that this was done (by others) using MODIS? Clarification is needed here what was done in the manuscript vs. prior work. Is it fair to say that the manuscript got closer to observational evidence for the validity of the interchangeability of spatial statistics and temporal evolution?

p15,L13: This seems like a fair statement, but how do we interpret it? For which purposes is the satellite-based method good enough, and for which problems do we

need to use airborne or tower-based observations (as later suggested by the authors)?

Minor comments:

p2,L17: 1) Why is there a new paragraph? 2) Suggest re-wording "In particular..." as "The phase transition ... is especially relevant for..." without a preceding indent/new paragraph.

p2,L25: remains > remain

p4,L30: "In further development of the scanning..." Something wrong with the language here and the conclusion of this sentence.p4,L12: add comma after "September)"

p4,L21: The "degree" characters should be superscripts.

p5,L7: "by measuring monochromatic radiation from a monochromator" - revise language?

p8,l17: "inlay" > "inset"?

p9,l31 (top of page) " grid cell in" > "grid cell at"

p9,l2: ranging between > from ... to?

p9,l3: What is the "first cloud case"? At this point, the table suggested above would really be helpful.

p9,l6: "originated" > "originating"?

p9,l8: The phase index is significantly shifted to positive values —> either it assumes positive values or not - what is the meaning of "significantly shifted to positive values" How about "shifted to positive values"?

p9,l9: Why "obviously"? Perhaps "apparently"? Meaning unclear.

p9,l14: "related" > "relative"?

p9,l15: move "also" to after "is"

p9,l20-21: Too hard to understand. Try to improve language.

p9,l22: "as can be concluded" > how can this be concluded?

p10,l1: "showing" > "with"

p10,l2: "need to be taken" > move to right after "images" on l1.

p10,l2: explain "epipolar plane"

p10,l3: What's the "world" coordinate system?

p10,l6: Usage of the word "exemplarily" seems out of place throughout most of the manuscript. How about "for example"?

p10,l6-7: Unclear what this means - wrong axis perhaps?

p11,L25: 6b: are the distances in km?

p11,L27: "quite *a* homogeneous" (add "a")

p11,l14: "scientific" > "science"

p11,l16: "which" > "because it" ?

p12,L30 (top): "have been" > "were"

p12,l32 (top): "phase states. Mainly" > "phase states, mainly"

p15,L15: ATTO - introduce acronym somewhere.
* * *

---

## Referee Comment (RC3) · Anonymous Referee #3 · 11 Mar 2017

General comments

Aerosols may alter cold-rain evolution leading to an invigoration of the DCC development. This paper presents observational results of vertical profiles of the cloud particle phase state in tropical deep-convective clouds (DCCs) using airborne solar radiation data taken in a field campaign held in Amazon. Spectral signature in cloud side reflection in the spectrum of wavelengths between 1.55 and 1.7 $\mu$m is used to deduce the mix phase zone in DCC. The altitude assignment is done by stereo-photogrammetric measurement using a commercial camera onboard the aircraft. Results show that depth

and vertical position of the mixed phase layer can vary by different stages of cloud development. The lower and upper boundaries of the mixed phase layer are higher in polluted aerosol conditions than in moderate conditions, although the number of samples is limited to draw a concrete conclusion. Observational evidence for shift of the liquid-to-ice phase transition zone is a key for better understanding of the aerosol effect. I think this paper is an important contribution even with a limited number of samples. The manuscript is generally well written. Results are clearly presented. I recommend that this paper is published with minor revisions. There are several suggestions for revisions as described below.

Specific comments

P4, L14: Is humidity variation small for selected 14 flight cases? If not, influence on the conclusions of this paper should be discussed.

Subsection 2.2.2 (MODIS) should be moved to after Subsection 2.2.4 (CAS-DPOL...) because Subsections 2.2.1, 2.2.3 and 2.2.4 describe the aircraft measurements.

P5, L25: The MODIS thermodynamic phase algorithm should be explained in more detail for better discussion (around P12, bottom) on the comparison of aircraft measurement results with the MODIS phase results.

P6, L24, "The aspherical fraction is the ratio of aspherical particles...": Is this a ratio of number concentration? Other definitions (area, volume, or mass) of the ratio are possible. Please clarify the definition.

P14, L4, the last sentence of this Subsection, "Also strong downdrafts can...": The last part of this sentence, "whereas in situ measurements inside the cloud only reveal liquid phase particles", is confusing to me. Why do cloud side observation and in situ measurements show different results?

Typographic corrections

P9, L32, "140 x 40 x 99": The "x" character should be replaced by the times symbol.

P9, L20, "get": Should be replaced by "become" or something.

P12, L22, "m s–1": Make the "–1" superscript.

---

## Author Comment (AC1) · 9 Jun 2017

**Reply to Reviewer #1:**

We thank the reviewer for the time and efforts she/he spent reading our manuscript and providing valuable suggestions and advices. Please find below a discussion of the reviewer's comments (italic). Changes/additions made to the text are underlined and given in quotes.

 *- The combination of geometric retrievals and hyperspectral measurements is new and can give additional cloud structure information. In addition its an important information to analyze angle dependent reflectance properties of the DCC relative to the Sun, but the derived parameter of cloud distance from the geometric retrieval is only one single parameter. The observed area by the imaging instrument depends on the field of view, cloud structure and distance and could lead to a 3D cloud structure, but the simple assumption of a homogenous vertical cloud area within the field of view of a single imaging pixel might lead to errors in the analysis of cloud scattering effects.*

The reviewer is right, that we used a simplified assumption in this work. A complete cloud structure retrieval needs much more efforts. In an upcoming publication (Zinner et al.) such a retrieval based on stereographic methods and the signature within the O2-A band will be presented. However, 3D effects due to horizontal photon transport is reduced at the wavelengths which are used for calculation of the phase index. The free photon path length is reduced due to cloud particle absorption in this spectral range. Marshak et al. (2006) and Martins et al. (2011) discussed the usage of 1D radiative transfer simulations for calculation of reflected radiation from cloud sides at different wavelengths. They concluded, that after adaption of the viewing and zenith angle,  1D assumptions are applicable for wavelengths, where cloud water absorption gets relevant and clouds have an optical thickness larger than 40 (Marshak et al., 2006), which is mostly fulfilled for DCCs. However, for the retrieval of the cloud particle radius the cloud structure gets more important due to a much higher contribution of scattering events.

*- A cloud masking procedure is introduced to distinguish between directly reflected areas of the cloud and diffuse shadow regions. The analysis of the manuscript is restricted to directly reflected areas only, which in fact is a sum of direct and diffuse light.*
*- Why does the described method of the distribution of the cloud phase does work only for direct reflected light of the Sun?*
*- What is the influence of the diffuse light?*

As the three comments above relate to each other, we give a joint response on them.
The reviewer makes good point here. In fact it is better to use the phrase "illuminated cloud regions" than talking about directly reflected areas. Of course, the measured reflected radiation contributes both, directly reflected radiation but also diffuse (multiple-scattering) radiation coming from other directions which fall into the sensor viewing angle. Since we are using radiation with wavelengths in the near infrared, the contribution of the diffuse radiation which is in-scattered from other directions is less dominant. For spectral radiation with wavelengths which are affected by Rayleigh scattering this in-scattering would be more relevant.

We added some more explaining sentences in the beginning of the section:

Compared to illuminated cloud sides, the photon paths in shadowed cloud regions are longer, which is related to more absorption events. This absorption due to cloud particles is not locally restricted to the cloud side parts where the camera is pointed at. In fact, the spectral radiation coming from shadowed cloud regions is affected by absorption by cloud particles from cloud parts outside the FOV of each individual spatial camera pixel. Since the spectral signature of

reflected radiation from shadowed regions of cloud sides is contaminated by a significant fraction of diffuse radiation originating from unknown cloud regions, a cloud masking technique was developed to discriminate illuminated and shadowed cloud regions.

For illustration of the effect of shadowed cloud parts we plotted the phase index of a cloud scene without cloud masking below. Flight altitude was about 4 km. All clouds shown here are liquid water clouds, because air temperature is higher than 0°C for this altitude range. The RGB image in the upper panel illustrate the position of the clouds, the lower panel displays the phase index of the same image section. The shadowed cloud regions show a phase index larger than 0.2, which would indicate the presence of ice particles. This illustrates clearly that shadowed cloud regions should be excluded from the data set, because the typical spectral signature of liquid and water clouds is lost.

[Figure]

*- It would be nice to see some more direct and detailed comparisons to the methods of Marshak (2006, reference missing) or Zinner (2008), MODIS, possibly Cloudsat and insitu. The description of Figure 9 could be in much more detail and as the major part from my point of view this is worth more than just one page.*

We surely could use several other satellite observation products for comparing our results, but we limited the measurement strategy to in situ (three instruments already) and the MODIS observations (ensemble method which was already applied for similar studies). The phase retrieval as presented by Marshak et al. (2006) (it's cited now) and Zinner et al. (2008) rely on the same approach. As shown in the reply on comment to Page 3 line 17, there is just a difference of the phase index in absolute numbers, but not in the height of the mixed phase layer itself (see directly reply to Page3 line 17). Furthermore, we modified the application section at various points. The main changes are given below.
When introducing the in situ measurements, we added the following sentences to specify the ideas behind the comparison of the different observation strategies:

The variability of the mixed phase layer in depth and height within a single cloud cluster shows that the vertical distribution at least at the cloud edges is variable. In situ data are used to investigate if such a variability is also observed in the more inner part of the cloud.

Later we added the following:

Furthermore, a second but smaller peak of the particle size was found at about 6 km altitude. From the conceptual model of cloud particle size profiles inside a DCC (e.g., Rosenfeld and Woodley (2003)) it might indicate the bottom of the mixed phase layer, when cloud particle size starts to increase. However, this increase is less pronounced than presented in Rosenfeld and Woodley (2003).

At the end of the subsection we added:

This shows that the satellite-based ensemble method may be representative for a large cloud field. But for individual clouds NIXE-CAPS and specMACS measurements have shown lower glaciation heights. The most likely reason is related to the fact that the ensemble method relies on cloud top observations of growing clouds in different stages of evolution. As shown in Fig. 9g mainly particle sizes between 22 and 27 μm were derived indicating that the profile is dominated by measurements of clouds in the mature stage. At this stage, the particle phase may be altered by up- and downdrafts within the clouds as was shown in Fig. 9e. This leads to an enhanced horizontal variability of the cloud phase state which cannot be resolved by passive remote sensing from cloud top observations. Another, but minor reason of the discrepancy between ensemble method and NIXE-CAPS / specMACS measurements is related to the retrieval uncertainty of the effective cloud particle radius. While scattering properties are well defined for liquid water particles, they are variable for ice particles due to differing habits and crystal shapes (Eichler et al., 2009). This gets even more complicated for cloud tops where phase transition starts. Additional retrieval uncertainties of the particle size directly contribute to the derived profile of $r_{eff}$.

*Page 2 line 5: A mixed phase state of water does not exist. I would rather describe it as an area of phase transition levels from existing state phases eg. from liquid to ice which can vary in temperature gradient, altitude and vertical depth (line 17 is here more precise than 5)*

That is a good point made by the reviewer. We changed the sentence as follows:

DCCs exhibit a high variability of cloud particle sizes and a complex vertical microphysical structure. This includes the different phase states of water (liquid and ice) of the cloud particles and the occurrence of layers where phase transitions between liquid water and ice particles (further referred to as mixed phase) take place.

*Page 2 line 13: ... (more aerosol particles ...)*

Changed into:

more aerosol particles

*Page 3 line 17: Why did Martins and Marshak use a different Wavelength? With SpecMACS it could be used as well and compared.*

The choice of wavelength pair is originated from the method described by Jäkel et al., (2013) which was designed for spectrometers not measuring at 2.1 and 2.25 μm wavelength. There are

several methods and wavelength ranges used and discussed in the past as listed in the introduction. The change of the sign of the phase index as calculated in this manuscript between liquid and ice phase is kind of illustrative. However, we compared the phase index profile for one of the cases derived by both methods as can be shown in the plot below. Apart from the absolute numbers we see a similar effect within the transition zone with a distinctive slope of both phase indices. So we don't see additional information when using 2.1/2.25 µm instead of radiances between 1550 and 1700 nm. The physics behind for both methods is similar; where the imaginary part of the refractive index which determines the spectral absorption is different between ice and liquid water particles in these two wavelength ranges. So it is not surprising, that the vertical profiles show the indication of the phase transition zone in the same vertical levels.

[Figure]

*Page 4 line 14:  Temperature profiles are mentioned and it would be very helpful to have some graphs. Sidewards looking IR-camera would be interesting.*

We tried to reduce the number of figures. Therefore, we omitted an individual plot showing the temperature profiles which are similar, as also stated in the overview paper by Wendisch et al., (2016). However, the relations between altitude and temperature can be estimated from the secondary y-axis showing the temperature as vertical coordinate in Fig. 9a for AC13 and in Fig. 10a,b for AC10 and AC18.
The reviewer makes a good point here to bring up the usage of an IR camera. Unfortunately, we had no IR-camera available for this campaign. But another ground-based campaign is scheduled for September/October in 2017 in the Brazilian rainforest using IR-camera and imaging spectrometers together for cloud side observations.

*Page 5 line 2: Again a comparison to the method of Marshak, ... is possible with SpecMACS.*

Please see answer on comment concerning Page 3 line 17.

*Page 5 line 29: mixed phase layer...→ phase transition layer*

The "term mixed" phase is commonly used in literature. We are aware that there is no mixed phase state as already commented by the reviewer. But we think the term "mixed phase" is not misleading here. For simplicity and consistency with literature we defined the layers of phase transition between liquid and ice particles as mixed phase in the introduction.

*Page 5 line 29:  A retrieval of cloud particle size of the measurements would demonstrate this sentence*

The retrieval of the cloud particle size from cloud side observations needs a lot of efforts and will be presented in a different publication entitled: "How accurately can we remotely sense cloud vertical profiles of droplet radius and phase from the cloud side perspective?" which is in preparation by Ewald et al.

We added a citation here where profiles of particle sizes together with estimations of the phase are presented:

The mixed phase layer is characterized by a strong increase of cloud particle size with height (Martins et al., 2011), whereas for fully glaciated cloud layers the largest ice particles can be found directly at the height where the glaciation temperature is reached.

*Page 6 line 11:  Please explain this statement, if its true.  Please compare state of polarization with Mie-Theory.*

*The* scattered and incident intensities of the polarization components are related by the phase function. This phase function is simplified for spherical particles due to their particle symmetry. We will not show the matrix operations here but we added a reference here which describes also mathematically the polarization of aspherical and spherical particles as measured with NIXE-CAPS:

Spherical particles do not strongly alter the polarization state of the incident light as discussed in detail by (Meyer, 2012), while non-spherical ice crystals change the polarization depending on their size and orientation (Nicolet et al., 2007; Meyer, 2012).

*Page 6 line 23 and 30: detection limit unclear. >1 cm-3 or 0.3 g/m-3*

These data and detection limits are from two different instruments integrated in the CAS-DPOL. We use the laser spectrometer on the CAS-DPOL (Cloud and Aerosol Spectrometer) to derive the aspherical particle fraction. Here, the cloud data are given for the size range between 3 and 50 µm and for clouds with a cloud particle number density > 1 cm-3. In addition, the hotwire instrument on the CAS-DPOL measures the liquid water content with a (conservative) detection limit of 0.3 g/cm-3.

*Page 7 line 3: Please explain the adjustment of the temperature, humidity, ... profiles*

In particular, the density of water vapor was re-calculated from measured temperature, pressure and relative humidity for each model height level. For simplifications we will not give the equations as they can be read in textbooks.

However, we adapted the sentences as follows:

For the model input, the atmospheric profiles of temperature, atmospheric pressure, and gas densities are taken from Anderson et al., (1986). From a radio sounding from Alta Floresta (-9.866° S, -56.105° W) and measurements of temperature, humidity and pressure performed by HALO, the temperature and pressure profiles are adjusted to represent the atmospheric conditions on 19 September 2014 (AC13) in the region of one of the measurement flights

(representative of the three flights considered in this study). The density of water vapor is re-calculated using the relative humidity, temperature and pressure measurements.

*Page 7 line 11: adjustment of the aerosol profile?*

The standard Shettle profile was scaled by the vertically integrated AERONET measurements. We are aware that this adjusted profile is just a rough estimate of the true vertical profile, but it will serve as input for radiative transfer simulations for sensitivity tests concerning cloud microphysical properties. As AOD decreases with wavelength, the aerosol extinction in the wavelength range between 1550 and 1700 nm is less important than cloud particle extinction in this spectral range.
We replaced "adjusted" by "scaled":

For the polluted case, aerosol properties are described with the model by Shettle (1989) and scaled by AERONET (AErosol RObotic NETwork) measurements (site Alta Floresta) of aerosol optical depth, single scattering albedo, and asymmetry parameter (used for the Henyey-Greenstein phase function).

*Page 7 line 19: As mentioned before. Why is the diffuse light restricted to shadow regions or does it have the same amount in the other regions as well?*

Please look for response on comments 2-4 above, since it deals with the same topic.

*Page 7 line 21: Here we have some weak indications why the diffuse light in shadow regions is not used in this study. While a thoroughly radiative transfer simulation can include the influence of ground reflectance, surface albedo in this manuscript can't be taken into account because of this influence. Why is that and a view sentences later the influence of the surface can't be seen in the airborne data? The reasoning in this part of the manuscript is somehow very weak.*

The reason for excluding the shadowed cloud regions from data evaluation is given above (see comments 2-4). It is a good question, why the effect of surface reflection was not observed during the aircraft measurements. Compared to ground-based observations, where the spectral features of the surface albedo (here vegetation step around 700 nm) can be found in the spectra of the reflected radiation of shadowed cloud regions (see left panel of the figure below from Jäkel et al., 2013), the cloud and observation geometry is different for the aircraft measurements during ACRIDICON-CHUVA. This is related to changes of the range of scattering angles, because reflected radiation is observed from higher altitudes than from the ground. Furthermore, for deep convective clouds the distance between surface and upper parts of the clouds is enhanced, which reduces the contribution of radiation coming from the surface. However, a detailed model study would be needed to quantify the surface effects on the reflected radiation coming from shadowed cloud regions to estimate the measurement conditions when significant spectral features can be used for shadow detection. Since we didn't observe such features, such a study will not be included in this work. In the figure below, the right panel shows clearly no indication of the vegetation step in the shadowed cloud region, while the surface observation shows the typical increase of radiation above 700 nm wavelength.

We modified this part as follows:
In ground-based observations the reflected radiation measured from shadowed cloud regions showed spectral signatures influenced by the spectral surface albedo due to interaction between clouds and the surface (Jäkel et al., 2013). This interaction is reduced for several reasons for aircraft observation of DCC. The reflected radiation is observed from higher altitudes than from the ground. This is related to changes of the range of scattering angles. Furthermore, the

distances between surface and in particular the upper parts of the cloud are much larger. Therefore, scattered radiation from the immediately adjacent cloud regions has a greater effect on the spectral features in the shadowed cloud areas than the surface. Since spectral indication of the surface could neither be observed nor simulated for airborne measurements, a different approach is chosen based on the distribution of color values in the observed cloud scene.

[Figure]

*Page 8 line 29: Where does this formula and the constants come from? I would propose to use a spectrum to rgb conversion via a CIE 1931 color space. SpecMACS has a broad spectral range and a large number of spectral channels why not using them? This would reduce noise as well.*

The equation calculates the relative luminance (CIE, 1999). We added the reference:

.., which takes into account the sensitivity of the human eye on the different colors by differential weighting of the three wavelengths (IEC, 1999)

IEC: Multimedia Systems and Equipment – Colour Measurements and Management – Part 2-1: Colour Management – Default RGB Color Space – sRGB, IEC 61966-2-1, International Electrotechnical Commission: Geneva, Switzerland, 1999.

We omitted the usage of "relative luminance" in the manuscript because it is a photometric quantity used in digital image processing and less known in the field of atmospheric science. As explained by Magisa et al. (2005), the "relative luminance (RL) is the relative brightness of any point in a color-space, normalized to 0 for the darkest black and 1 for the brightest white. For a certain point (or pixel) in a color image encoded in the standard RGB (sRGB) color-space, the RL can be computed based on the value of the sRGB components through the equation $RL = 0.2126R + 0.7152G + 0.0722B$".

RGB conversion via CIA 1931 uses a spectral weighting of the red, green, and blue channels, where the weighting function corresponds to the spectral response of the human eye. The color matching function as taken from CIE is shown below:

[Figure]

The spectral bandwidth of the three weighting functions is quiet broad. If there is a difference in the spectral signature of the radiances between shadowed and illuminated cloud areas, then the usage of CIE bands is not recommended, due to the loss of spectral information after applying the spectral convolution. However, using RGB values to classify the brightness of the individual pixels doesn't require the full spectral information as it could be provided by specMACS. The simple approach to identify the directly illuminated cloud areas based on the three wavelengths (436, 555, 700 nm), has been approved by the simulations.

But for the upcoming ground-based campaign in September/October 2017 we will use the whole spectral information from 400 – 2500 nm to gather information on the illumination conditions as found in Jäkel et al. (2013).

*Page 8 line 1, ...:  How is the histogram of the RGB values converted or evaluated to the frequency distribution? Please explain in more detail. Where does the relative and absolute frequency come from in fig. 3? Why are the simulated once in Fig 3b absolute and the measured once in Fig 3c relative? The calculation of a single RGB value with the formula is used to find the threshold of "directly !" illuminated pixels. What are the model simulations for if you dont use them?*

The RGB histogram (= frequency distribution) derived from the simulations was shown to illustrate that such histograms can be used to discriminate between the illuminated and shadowed cloud regions. The threshold estimated from the distribution of the RGB values is just an example and not valid for other cloud scenes with different observation geometry. But we see clearly that the modes in the histogram match with the illuminated and shadowed cloud regions as classified from the known geometry in the model. We plotted now both histograms (from simulation and from measurements) as relative frequency as suggested by the reviewer:

[Figure]

Furthermore, we modified the text:

The histogram of the RGB color values for each cloud scene is used to identify the illuminated and shadowed cloud areas. Before showing an application, the procedure is illustrated using simulated cloud side reflectivity observations. In this manner, we can directly compare the classification of illuminated and shadowed cloud regions (i) derived from known cloud and viewing geometry, and (ii) derived from the histogram of the RGB color values.

And later:

The histogram of the simulated RGB color values is shown in Fig. 3b as black line. Two modes are visible, which coincide with the two sub-classes of illuminated (red) and shadowed (blue) cloud regions as calculated from the cloud and viewing geometry.

*Page 8 line 3: What is the max height of the model domain?*

The maximum height was 120 km corresponding to the top of atmosphere. We added the top height and vertical resolution to the text:

The cloud field was generated by the Goddard Cumulus Ensemble model (Tao et al., 2003, Zinner et al., 2008) for a model domain of 64 x 64 km with a horizontal resolution of 250 m and a vertical resolution between 0 and 10 km altitude of 200 m. From 10 to 120 km altitude the simulations are performed with a vertical resolution ranging between 1 and 5 km. The maximum extension of the liquid water clouds from bottom to cloud top ranges from 1.0 to 7.4 km altitude.

*Page 8 line 16: What is a relative azimuth angle of exactly 68 degree with a changing attitude and Sun elevation during airborne missions?*

The data given here are valid for the cloud scene (about one minute of flight with constant heading) which is shown in the Figure 3. Of course the distribution of RGB color values has to be calculated for each cloud scene separately. It is not meant here, that this histogram and the related threshold is valid for the entire flight. In fact, the thresholds depend on the illumination conditions and viewing geometry. We modified the section to make it clearer for the reader.

The procedure is applied exemplarily for a cloud scene observed during ACRIDICON-CHUVA from 19 September 2014. During the roughly one minute flight leg the aircraft did not change its flight attitude, resulting in almost constant relative azimuth angle (angle between the sun and the viewing direction of specMACS) of 68° and solar zenith angle (theta = 39°). Note, that all other selected cloud cases in this study have similar restrictions concerning the flight attitude and time period (about one minute) to guarantee comparable illumination conditions in one cloud scene. Fig. 3c illustrates the RGB histogram as calculated for observations of specMACS with an elevation ranging between -13 and +12°.

*Page 8 line 22,23: The simulation shows an increase in cloud particle size in Fig 4 for that region. What is wrong?*

In the beginning of Section 3.2 a short motivation is given why a phase index may be a better indicator for the location of the transition layer than using the vertical profile of the cloud effective radius as used by Rosenfeld and Woodley (2003). But as mentioned, there are cases where the particle radius doesn't increase with decreasing temperature. For this reason, we used the phase index.
Fig. 4b shows one example of a profile with variable effective radius and water content. There was no intention to derive the profile of the phase index typical for marine, continental and polluted conditions. We restricted the simulations to two special cases, first, a constant distribution of Reff and LWC/IWC with height, where the effect of variations in the microphysical properties (apart from the particle phase) on the phase index can be neglected and second, a typical cloud profile with variable microphysics.

Furthermore, modified the motivation for showing additional radiative transfer simulations:

In the following, results from radiative transfer simulations using MCARATS are presented. The viewing geometry and the atmospheric description are adapted to the conditions during ACRIDICON-CHUVA on 19 September 2014. These simulations are performed to demonstrate that ice and liquid water phase can be separated from the transition layer under different conditions similar to the results reported by Jäkel_et al. (2013). Note, that due to the different viewing geometry, another angular range of the scattering phase function is observed than for ground-based measurements. This might have an effect on the characteristics of phase index profile in particular with respect to separation of the mixed phase layer.

And later the two cloud scenarios are introduced as follows:

Two simplified cloud scenarios with different profiles of cloud effective radius and water content are assumed. In both cases the clouds ranged from 4.0 to 11.0 km altitude with a mixed phase layer between 6.4 and 7.0 km. While the first scenario uses constant values of cloud effective radius ($r_{eff}$ = 20 μm for liquid water and ice) and water content (0.7 g m$^{-3}$), the second scenario assumes variable profiles of the microphysical parameters. These two cases are chosen to identify effects on the $I_P$-profile caused by changes of (i) the phase state itself (scenario 1), and changes of (ii) the cloud particle size and water content (scenario 2).

*Page 8 line 28: How does this simple formula compare to the methods from Marshak, Martins and Zinner?*

We gave some additional information:

For ground-based application with corresponding viewing geometry vertical profiles of the phase index were simulated by Jäkel et al. (2013). A significant gradient in the vertical profile of the phase index was observed between liquid water and mixed phase layer, but also between mixed phase layer and ice phase. A similar behavior was also found for the reflectance ratio at 2.10 and 2.25 μm as reported by Martins_et al. (2011). They observed a strong gradient in the profile of the reflectance ratio. This is due to the fact, that the imaginary part of the refractive index, which determines the spectral absorption, is different between ice and liquid water particles in the two wavelength ranges used by \Martins_et al. (2011) and Jäkel et al. (2013).

*Page 9 line 5: Is the combined Ip profile a simulated or measured profile. I don't understand how the combined profile is calculated and where it comes from.*

We modified the sentences as follows:

From the 3D simulations of the spectral radiance at 1550 and 1700 nm the phase index is calculated following Eq. (2). For each modeled grid cell in the model domain with a horizontal distance between 3 and 8 km to the cloud, a combined $I_P$-profile is derived from the different viewing elevation angles. Such $I_P$-profiles are plotted in Fig. 4a in black dots.

*Page 9 line 7: three phases ?*

We changed the sentence, also later in line 11.

For the first scenario with constant microphysical parameters, three distinct clusters corresponding to the phase state of water and the zone of phase transition, with negative values for pure liquid water, can be found.

And:

The variability of the phase index for constant microphysical conditions in each of the phases is caused by the effect of the different viewing geometries.

*Page 9 line 10: What is a pronounced absorption, of what?*

Changed as follows:

This might be caused by the fact that the contribution of ice particles within the mixed phase layer leads to an increased absorption of radiation resulting in an increase of the phase index.

*Page 9 line 12: Each cloud height → The cloud vertical structure is ...*

*Changed as suggested:*

The vertical cloud structure is observed from different sensor elevation angles and distances.

*Page 9 line 14: To derive the particle size is first mentioned here. Is that the goal or what is the reason? A look up table would do as well, please look at AMT Zinner 2016.*

It seems that this sentence is misleading. Therefore, we deleted it from the manuscript. The retrieval of the effective radius is not object of this work.

*Page 9 line 15: What is a more realistic cloud? Are the other clouds not realistic?*

We modified this part as follows:

The second cloud scenario assumes variable cloud microphysical properties. In general, in convective clouds, the size of ice particles is higher than the size of liquid water particles. Therefore, the second scenario represents a more realistic vertical distribution of the particle effective radius and water content than the first scenario.

*Page 9 line 17: What is the first case?*

We better introduced now the two cloud setups as used for the radiative transfer simulations and omitted the phrase "case" in this section. See also reply on comment Page 8 line 22,23.

*Page 9 line 18: The transition layer is characterized by a strong increase in particle size and change in the value of phase index. See Fig 4b (simulations) and Fig. 8*

The sentence refers to the description of the microphysical parameters as illustrated in Fig. 4b. Therefore, no information about the phase index is given here. It follows some lines below.

*Page 9 line 24: I assume that we have a polluted and a clear case, but it's not clear in this part of the manuscript. Here we have only two cloud cases, one with fixed microphysics and one with changing cloud properties. Please clarify.*

The two scenarios shown here are intended to demonstrate if the phase index can resolve the three layers with viewing geometry from the aircraft observations. So, we haven't chosen the two scenarios with respect to aerosol conditions. It should be getting clearer for the reader after modification of the beginning of the section (see reply on comment Page 8 line 22,23:)

*Page 10: Geometry is Ok, but could be shorter. Except a real 3D cloud structure would be the final product.*

Another publication is in preparation for AMT which will discuss the 3D reconstruction of clouds based on photogrammetry and O2-A band absorption (Zinner et al.).

*Page 12 line 12: A profile and comparison of remote sensing and insitu droplet size would be interesting. A sharp transient of the droplet size shows the transition layer.*

Indeed, a profile of the in situ measured particle size would be interesting. But in situ measurements have the disadvantage that they provide only data along the flight path. As we see from the satellite picture, a large cluster was probed during AC13. The flight altitude is color coded in the right panel (see plot below). From this flight pattern no profile of a single cloud is available, because the flight altitude varied over a large area comprising different clouds of different evolution stages in the cluster.
A combined profile of the effective particle diameter is shown below. The data are based on measurements of the CAS-DPOL and CIPg (Cloud Imaging Probe grayscale, size range: 15 to 960 µm, operated by Mainz University). A distinct increase of the particle size cannot be observed, neither by the CIPg, nor by the CAS-DPOL (size range < 50 µm).

[Figure]

*Page 12 line 30: mixed phase levels → phase transition levels or better layer*

*Fully developed deep convective clouds with cloud tops between 10 and 14 km (classified as ice cloud) and low level cumulus clouds up to 6 km (liquid water clouds) are detected. Cloud phase information from the assumed phase transition layers is not available in Collection 6.*

*Page 13 line 20: Why are liquid water data from up to 8.7 km not shown*

We didn't show the time series of the NIXE-CAPS data for AC18 as a separate plot as provided for AC13. The phrase "not shown" is removed from the text. In case the reviewer is interested in the time series, please find the plot of the data below:

[Figure]

*Page 14 line 2: A temp profile is missing.*

Fig. 10 also includes a secondary y axis illustrating the temperature as vertical coordinate.

*Page 14 line 25: three phases?*

The sentence was changed as follows:

Depending on the viewing geometry and cloud distance, layers of pure liquid and ice phase, as well as phase transition layers were identified.

*Page 14 line 29: Is there only one polluted case during the whole campaign?*

Cecchini et al., (2017) have listed the characteristics of the flights illustrating the aerosol conditions:

**Table 1:** General characteristics of the cloud profiling missions of interest to this study: condensation nuclei ($N_{CN}$) and CCN concentrations ($N_{CCN}$, with $S = 0.48\% \pm 0.033\%$), cloud base and 0 °C isotherm altitude ($H_{base}$ and $H_{0°C}$, respectively), start and end time and total number of DSDs collected. The data are limited to the lower 6 km of the clouds. The unit for $N_{CN}$ and $N_{CCN}$ is $cm^{-3}$ and the unit for altitudes is in m. Profile start and end are given in local time.

| Region | Flight | $N_{CN}$ (cm⁻³) | $N_{CCN}$ (cm⁻³) | $H_{base}$ (m) | $H_{0°C}$ (m) | Start | End | # DSDs |
|---|---|---|---|---|---|---|---|---|
| Atlantic Coast | AC19 | 465 | 119 | 550 | 4651 | 13:17 | 14:57 | 630 |
| Remote | AC09 | 821 | 372 | 1125 | 4823 | 11:30 | 14:21 | 665 |
| Amazon | AC18 | 744 | 408 | 1650 | 4757 | 12:32 | 14:14 | 397 |
| Arc of Deforestation | AC07 | 2498 | 1579 | 1850 | 4848 | 13:49 | 17:16 | 674 |
| | AC12 | 3057 | 2017 | 2140 | 4938 | 12:55 | 15:16 | 381 |
| | AC13 | 4093 | 2263 | 2135 | 4865 | 12:46 | 15:36 | 204 |

AC13 was the most promising flight to measure polluted conditions with the largest number of condensation nuclei. For AC12 most of the flight was performed at flight altitude below 6 km, therefore no deep convective clouds have been observed by specMACS.

*Page 14 line 30 bottom: Low statistics? Are those 2 flights analysed in this study the only possible ones of the whole campaign?*

From the 14 scientific flights we selected the three days (AC10, AC13, and AC18) with the best conditions as stated in beginning of Sect. 4:

(i) no cloud layer above the observed cloud (no cirrus), which contaminates the spectral signature,
(ii) high proportion of illuminated cloud parts in the vertical direction of the cloud,
(iii) flight altitude that allows measurements of an extended vertical region of the cloud considering the limited FOV of specMACS, and
(iv) isolated clouds with recognizable structures for cloud geometry retrievals.

This limits the number of cases. Similar limitations are also reported for the in situ data sampling as shown in Costa et al., (2017). They had data from cloud passages lasting between 1 and 18 minutes in sum per flight.

---

## Author Comment (AC2) · 9 Jun 2017

**Reply to Reviewer #2:**

We thank the reviewer for the time and efforts she/he spent reading our manuscript and providing valuable suggestions and advices. Please find below a discussion of the reviewer's comments (italic). Changes/additions made to the text are underlined and given in quotes.
The general comments made by the reviewer summarize the main points of the specific comments and suggestions. Therefore, we will start with our replies on the specific and sequential comments.

*p3,L5: I don't see the relevance of the cited paper (Cahalan, 1994) and the associated science (plane-parallel retrieval assumptions) in this context.*

Since the profile retrieval of the cloud phase from MODIS as applied later is based on the cloud particle size retrieval, we included here also the limitations of the size retrieval with respect to the bias caused by 1D assumptions. However, we removed parts of the text and added the following:

From the ensemble of retrieved effective droplet sizes, a vertical profile of cloud phase can be estimated because of the relationship between cloud phase and vertical profile of the cloud particle size (Rosenfeld and Feingold, 2003; Yuan et al., 2010; Martins et al., 2011). However, the retrieval of the effective droplet size relies on one-dimensional (1D) radiative transfer simulations, which incorporates retrieval uncertainties due to plane-parallel cloud assumptions and neglecting the net horizontal radiative transport between the satellite pixels (Zinner et al., 2006). Consequently, a decrease of pixel size causes an increase of the independent pixel bias, because the smaller the pixel, the more important is the net horizontal photon transport, particularly for the wavelengths in the visible spectral range, which are used for the retrieval of the effective droplet radius.

*p3,L17+L30: These are places where the manuscript could outline how the specific work fits into the larger context that was set up previously. Currently, this page in particular looks like a list of work done by the authors and predecessors (e.g., Martins, Rosenfeld), with only tangential connection to the motivation from the previous page(s). This is not a deal breaker for the manuscript, but it would be better to see how the listed work serves a number of outstanding questions related to the introductory comments earlier on.*

Starting from p2l25 a technical and experimental review is given how profiles of the cloud microphysical parameters can be derived. This review is mainly focused on passive remote sensing approaches, either from satellite observations (ensemble method), or aircraft observations but also from ground-based measurements. Data products based on these approaches will be used in this study. Therefore, it is worthwhile to describe them shortly here. Most of the cited literature is related to a technical description of the individual retrieval approaches with no or only less discussion on e.g., aerosol-cloud interaction.
To strengthened the connection between motivation and the presented phase retrieval we added the core questions at the end of the introductions:

In this paper we will address the following questions: (i) Can we observe differences in the vertical distribution of the thermodynamic phase state in DCCs for different aerosol conditions by using cloud side observations? (ii) How do the vertical profiles of cloud phase derived from cloud side observations agree with results from satellite (ensemble method) and in situ measurements?

*p4,L4: In the description of the manuscript's structure, it seems that the goal is to compare remote sensing derived cloud profiles to MODIS and in-situ data, which would make the manuscript more appropriate for AMT than for ACP. If, however, there are some higher-level goals that address some of the questions brought up above, this should be made clearer.*

The manuscript was submitted to be published in a special issue of AMT/ACP presenting results from the ACRIDICON-CHUVA campaign. Admittedly, parts of the manuscript include the description of the applied method, but compared to former publications dealing with cloud side observations, this manuscript presents the results of an entire campaign. We are aware, that the number of cases is limited. From the 14 scientific flights we selected the three days (AC10, AC13, and AC18) with the best conditions as stated in the beginning of Sect. 4:

(i)      no cloud layer above the observed cloud (no cirrus), which contaminates the spectral signature,

(ii)     high proportion of illuminated cloud parts in the vertical direction of the cloud,

(iii)    flight altitude that allows measurements of an extended vertical region of the cloud considering the limited FOV of specMACS, and

(iv)    isolated clouds with recognizable structures for cloud geometry retrievals.

This limits the number of cases. Similar limitations are also reported for the in situ data sampling as shown in Costa et al., (2017). They had data from cloud passages lasting between 1 and 18 minutes in sum per flight.

We adapted the outline of the paper to point out that the retrieval method is applied to different cloud cases under different aerosol conditions and discussed using also other observation strategies.

The variability of vertical phase distribution is discussed with respect to aerosol conditions and compared to in situ and MODIS products.

*p4,section 2.1: A table with flights and clouds cases would help.*
We added a table summarizing the three flights which are presented in this work.

A summary of the three flights used in this work is given in Table 1.

**Table 1.** Summary of presented flights with cloud side observations during the ACRIDICON–CHUVA campaign. The ranges of flight altitude and time refer to the studied cloud cases.

| Flight number | AC10 | AC13 | AC18 |
|---|---|---|---|
| Aerosol conditions | moderate | polluted | moderate |
| AOD (MODIS) | 0.4 - 0.5 | 0.5 - 0.6 | 0.3 - 0.4 |
| Number of cloud cases | 9 | 16 | 10 |
| Flight altitude range (km) | 7.4 - 10.4 | 5.2 - 9.3 | 1.4 - 14.0 |
| Time range (UTC) | 17:25 - 19:20 | 17:55 - 19:00 | 15:30 - 20:30 |

*p5,L12-18: The manuscript should elaborate on the stereo algorithm a little more. Also, L21, should this be "assign" instead of "allocate"?*

Please read our response to the commentary on p.10 dealing with the stereo algorithm. Furthermore we changed the sentence as suggested:

This allows assigning elevation and azimuthal angle to each point of the image.

*p6,L23: The description of the aspherical fraction is a bit unclear; what is measured, and what is derived?*

The aspherical fraction from the CAS-DPOL is determined by measuring the perpendicularly polarized light in the backward direction and the forward scattering light intensity. While the forward scattered light intensity is used to determine the size of the particle, the ratio of the forward and the backward scatter light determines the phase of the particle. While spherical particles do no change the polarization ratio, aspherical particles do. In order to categorize into liquid and ice particles, a size dependent threshold was inferred from calibration measurements of spherical liquid particles in the AIDA cloud chamber (Järvinen et al., 2016, Schnaiter et al., 2016). Particles with a polarization ratio larger than the 1- sigma range of the inferred sphericity-threshold were categorized as aspherical. The method gives a size dependent aspherical fraction of the first 300 particles measured each second. The bulk aspherical fraction was derived from the number of aspherical particles to the number of total particles measured between 3 and 50 μm per second.

We condensed it, because the CAS-DPOL principle is similar to that of the NIXE-CAS:

The aspherical fraction (AF) from the CAS-DPOL is determined by measuring the perpendicularly polarized light in the backward direction and the forward scattering light intensity. The ratio of the forward and the backward scattered light determines the phase of the particle. Particles with a polarization ratio larger than the 1-sigma range of the inferred sphericity-threshold are categorized as aspherical. The method gives a size dependent aspherical fraction of the first 300 particles measured each second. The bulk aspherical fraction is derived from the number of aspherical particles to the number of total particles measured between 3 and 50 μm per second.

*p6,L30: Why is the threshold this high? 0.3 g/m3 seems excessive, considering that a typical BL cloud top-level LWC is 1 g/m3.*

We added the following:

The Hotwire sometimes returns a signal in ice or clouds of partly frozen particles. This signal is on the order of 0.2 g m$^{-3}$. Thus a conservative threshold of 0.3 g m$^{-3}$ is used to reduce the false alarm rate.

*p7,l1: Why is it necessary to perform 3D calculations? Only because of the geometry, or because deviations from standard 1D models are expected? If so, what are they (aside from shadows)*

A convenient way to simulate cloud side reflections is to use 3D radiative transfer models, where the geometry of the observation strategy can be directly transferred to the model setup. There are ways to use 1D radiative transfer simulations instead by adapting the viewing and zenith angle. But this underlies restrictions, because no horizontal transport can be considered. This was shortly discussed in Marshak et al. (2006) and Martins et al. (2011). With respect to 3D radiative effects, less impact due to shorter photon paths is observed for absorbing wavelengths as were used for the phase retrieval. However, at cloud edges with lower optical thickness (tau < 30 after Martins et al., 2011) the cloud reflection at cloud particle absorbing wavelengths is still variable, but gets saturated starting from tau = 40 (Marshak et al., 2006). Summarized, with some limitations the phase indices could be also derived by plane parallel simulations. But in any case considering cloud shadow effects (as in section cloud masking procedure) 3D simulations were necessary.

*p7,l6: Rayleigh scattering → molecular scattering?*

We use the term "Rayleigh scattering" (as used in Bodhaine et al., 1999) because it describes the scattering on atoms and molecules.

*p9: Here it becomes quite difficult to understand what the authors are after - the position/width of the mixed layer? Is that the purpose of the simulations? Are they done to only replicate the earlier study, or are they something new?*

We hopefully introduced the purpose of the simulations better now:

In the following, results from radiative transfer simulations using MCARATS are presented. The viewing geometry and the atmospheric description were adapted to the conditions during ACRIDICON-CHUVA on 19 September 2014. These simulations were performed to demonstrate that ice and liquid water phase can be separated from the transition layer under different conditions similar to the results reported by Jäkel et al. (2013). Note, that due to the different viewing geometry, also a different angular range of the scattering phase function was observed than for ground-based measurements. This might have an effect on the characteristics of phase index profile in particular with respect to separation of the mixed phase layer.

*p10: The geometry retrieval description is rather cryptic. Would it help to cite work related to MISR, or is the method unrelated? What happens if the cloud moves during two consecutive images (used for the stereo method)? Also, elaborate on p10,l18-19, and state with respect to which coordinate system the "elevation angle" is provided.*

We omitted a detailed description of the method because it includes a lot of equations which are given elsewhere. But for better understanding we referred to a publication which discussed the mathematics for a similar experimental setup (Biter et a., 1983):

The theoretical background on photogrammetry is given in Hartley and Zisserman (2004), while Hu et al. (2009) applied these techniques for cloud geometrical reconstruction. The mathematics for the geometry retrieval, as it is used in this study, is based mainly on the method described by Biter et al. (1983). They deployed a side-looking camera onboard of an aircraft to detect the position of cloud features, similar to the setup presented in this work.

And later:

After coordinate transformation, trigonometric methods (Biter et al., 1983) are applied to calculate the distance between the camera positions P1 and P2 to the observed point C.

We added also a short comment on the meaning of the "elevation angle":

Repeating this procedure for a number of points yields a relation between elevation angle and cloud height. Note, that the elevation angle represents the elevation angle of the selected tie point of the camera image after correction based on the aircraft attitude data. It basically gives the elevation angle above or below the flight altitude.

We are aware that cloud movement might introduce an additional uncertainty. Therefore, we tried to reduce the time between two consecutive pictures. The GoPro delivered a movie, such that images from different time intervals can be selected for one cloud scene. All evaluated cloud scenes used data from time intervals less than 10 seconds.

The same tie points are chosen in a second image taken about 10 seconds later. Choosing a short time interval helps to reduce the uncertainty of the method induced by cloud movement.

Note, that another publication is in preparation for AMT which will discuss the 3D construction of clouds based on photogrammetry and O2-A band absorption (Zinner et al.).

*p10: Others circumvented the whole (rather difficult) stereo algorithm by including an IR channel. Why was that not done here? Were these measurements simply not available? And why did the authors prefer the more complicated method to the simple IR imager?*

The reviewer makes a good point here to bring up the usage of an IR camera. Unfortunately, we had no IR-camera available for this campaign. But another ground-based campaign is scheduled for September/October in 2017 in the Brazilian rainforest using IR-camera and imaging spectrometers together for cloud side observations.

*p11, Applications section. What is it that the paper seeks to find out? Refer to the main question here, at least at the beginning? This whole section reads a little bit like a listing of results with no specific purpose. Quite surely there is one, and that should be clarified more.*

We reordered the beginning of this section a little bit and introduced this sections with the two main questions of the case study:

From the 14 scientific flights three days (AC10, AC13, and AC18) are selected with the best observation conditions for specMACS, namely: (i)  no cloud layer above the observed cloud (no cirrus), which contaminates the spectral signature, (ii) high proportion of illuminated cloud parts in the vertical direction of the cloud, (iii) flight altitude that allows measurements of an extended vertical region of the cloud considering the limited FOV of specMACS, and (iv) isolated clouds with recognizable structures for cloud geometry retrievals.
Phase profiles from AC13 representing polluted aerosol conditions will be compared to the two days with less aerosol pollution. Effects of aerosol conditions on the height and thickness of the mixed phase layer will be investigated. Second, it will be demonstrated how comparable the different observation strategies (cloud side, cloud top and in situ) are.

We restructured the subsections a little bit to separate the two major goals of this sections. At the end of Sect. 4.1 we discussed the comparability of the different observation strategies, while at the end of Sect. 4.2 the aerosol impact on the mixed phase layer is summarized.

Sec. 4.1: see comment referring to *p13,L12-14*

Sec. 4.2:
From theory, the mixed phase layer is expected to be higher for polluted aerosol conditions than for cleaner aerosol conditions, which can partly be confirmed by comparison of the three cases. We found from cloud side observations, that the lower boundary altitude of the mixed phase layer tends to be higher for polluted conditions (AC13: 6.0 - 6.5 km) than for the moderate case of AC18 (5.6±0.2 km), while the upper boundary is shifted from 6.8±0.2 km (moderate case AC10) to 7.4±0.4 km (polluted case AC13).

*p12: It is a bit unclear what the in-situ measurements really have to offer here if "the direct comparison of in situ and remote measurements is difficult". Really, the in-situ measurements*

*should serve as validation for the remote sensing, but what do we learn if that doesn't work? Is it still worth using the in-situ data? If the comparison does not work out, what does that mean for the initial hypotheses (if there is one: perhaps a question about the interchangeability of pixel/time mentioned earlier?)*

When introducing the in situ measurements we added the following sentences to specify the ideas behind the comparison of the different observation strategies:

The variability of the mixed phase layer in depth and height within a single cloud cluster shows that the vertical distribution at least at the cloud edges is variable. In situ data are used to investigate if such a variability is also observed in the more inner part of the cloud.

*p13: The in-situ data puts ice higher than remote sensing. Which is right? What do we learn here about the representativeness of satellite data and \*its\* consistency with the aircraft measurements?*

We didn't find much differences between the glaciation heights derived from the ensemble method based on particles sizes from MODIS (9 km) and the estimations from the CAS-DPOL (8.7 km). Both are using a larger sample of data which were averaged over the entire cloud cluster. While the time series of the NIXE-CAPS instrument shows for individual cloud passages a glaciation height of 8.0 km which is in much better agreement with the cloud side observations. An assessment about the comparability is given at the end of this subsection (please see comment p13,L12-14).

*p13,L34 (top): So what phase \*does\* MODIS get for 6km? The shown results are certainly not liquid drops, given the large size.*

It's a good point made by the reviewer. We assume that this second peak might indicate the bottom of the mixed phase layer. We added the following:

From the conceptual model of cloud particle size profiles inside a DCC (e.g., Rosenfeld and Woodley, 2003) it might indicate the bottom of the mixed phase layer, when cloud particle size starts to increase. However, this increase is less pronounced than presented in Rosenfeld and Woodley (2003).

*p13,L12-14: Here we get some potentially important conclusions, which should be expanded an elaborated on. What is the significance of this finding? What can the satellite-based ensemble method do, and what not? Do in-situ and remote sensing observations from the aircraft tell two different stories?*

It's hard to give a general conclusion on the validity of the ensemble method due to the limitations of studied cases. For such a statement another study would be required using data from several measurement campaigns comparing in situ with satellite observations. However, for our data set we can conclude the following:

Comparing the glaciation height from MODIS with NIXE-CAPS in situ data and results from specMACS observations shows a deviation of about 1.0 - 1.5 km between the different retrieval techniques and observation strategies. However, the mean profile over the entire cloud cluster derived from CAS-DPOL measurements exhibited a similar glaciation height (of about 8.7 km) as found from the MODIS data. This shows that the satellite-based ensemble method may be representative for a large cloud field.

But for individual clouds NIXE-CAPS and specMACS measurements have shown lower glaciation heights. The most likely reason is related to the fact that the ensemble method relies on cloud top observations of growing clouds in different stages of evolution. As shown in Fig. 9g mainly particle sizes between 22 and 27 μm were derived indicating that profile is dominated by cloud measurements in mature stage. At this stage the particle phase may be altered by up- and downdrafts within the clouds as was shown in Fig. 9e. This leads to an enhanced horizontal variability of the cloud phase state which cannot be resolved by passive remote sensing from cloud top observations. Another, but minor reason of the discrepancy between ensemble method and NIXE-CAPS / specMACS measurements is related to the retrieval uncertainty of the effective cloud particle radius. While scattering properties are well defined for liquid water particles, they are variable for ice particles due to differing habits and crystal shapes (Eichler et al., 2009). This gets even more complicated for cloud tops where phase transition starts. Additional retrieval uncertainties of the particle size directly contribute to the derived profile of $r_{eff}$.

*p14,l5: Does this explain the discrepancy between in-situ obs and remote sensing?*

We added the following:

Also local strong downdrafts can transport ice particles into lower levels, which will be interpreted as mixed phase layer from the cloud side observation perspective. Due to the horizontal variability of cloud phase inside a cloud cluster for example caused by up- and downdrafts, in situ measurements may only reveal liquid phase particles. A direct comparison between the observation strategies is subject to restrictions because of temporal and spatial variability of cloud properties in convective systems.

*p14,l16/17: distinctive \*change\* in gradient, or simply "significant gradient" (change of gradient is a gradient of a gradient ... )*

Changed as suggested.

*p14,L21: Here the question is again why this was chosen over IR imaging.*

As mentioned already, an IR imager was not available during the campaign.

*p14,L31: Earlier, the authors said that the aircraft measurements are not statistically significant to prove/disprove theory. This statement here is not meant to be the main finding of the manuscript, is it? Wouldn't satellite data be more suitable to put this on a statistical basis? If so, what would then be the purpose of the aircraft measurements? This may be obvious, but it would help the reader to understand this point.*

In this part of the conclusions we summarized the findings of the comparison between the three flights (AC10, AC13 and AC18) with respect to aerosol impact on the mixed phase layer height. Conclusions about the comparability is given later (see reply to comment p15,L13).

For moderate aerosol conditions, only few cases exhibited liquid water, mixed phase, and ice phase, which limited the statistical significance of the comparison with AC13. However, comparing the glaciation heights of AC10 (6.8±0.2 km) and AC13 (7.4±0.4 km) we found an indication of an increase of glaciation height and a decrease of glaciation temperature for polluted aerosol conditions. With respect to the occurrence of first ice particles, the lower boundary of the mixed phase layer was derived with 6.0 - 6.5 km for polluted conditions, whereas for AC18 the altitude was shifted down to 5.5 - 6.0 km, which agrees with theory.

*p15,L38(top): The results from remote sensing and in-situ are not really consistent (as discussed earlier by the authors, and noted by the reviewer).*

This sentence was referring to the comparison between NIXE-CAPS and specMACS measurements. We changed the sentence for clarification:

Consistent results of mixed phase zone levels were found from specMACS and NIXE-CAPS measurements, for the flight AC13 with most individual cloud cases showing pure liquid, mixed phase layer and pure ice phase.

*p15,l3: "invariance of space and time" does not seem an appropriate way to describe the assumptions of the ensemble method. It's really spatial statistics vs. temporal evolution, isn't it? Secondly, the manuscript now divulges that it did seek to study the aerosol effect on deep convection - or is this a statement that this was done (by others) using MODIS? Clarification is needed here what was done in the manuscript vs. prior work. Is it fair to say that the manuscript got closer to observational evidence for the validity of the interchangeability of spatial statistics and temporal evolution?*

The reviewer is right when describing the ensemble method assumptions by spatial statistics vs. temporal evolution. But we will use here the term "time-space-exchangeability" as it was named in several other publications (e.g., Lensky & Rosenfeld, 2006, Yuan et al., 2010). The main intention of this is short paragraph was to summarize the results of the ensemble method. It was not intended to study aerosol effects on DCCs on the basis of cloud top observations. We rephrased the first sentence for clarification as follows:

Additionally to in situ and cloud side measurements, the glaciation temperature was derived applying an ensemble method based on MODIS data, which assumes time-space-exchangeability for a cluster of clouds with different states of evolution.

*p15,L13: This seems like a fair statement, but how do we interpret it? For which purposes is the satellite-based method good enough, and for which problems do we need to use airborne or tower-based observations (as later suggested by the authors)?*

We made a final conclusion on this as follows:

It is concluded that the assumed time--space--exchangeability used in the ensemble method can give a simplified picture of the vertical distribution of the phase within a field of convective clouds of different stages of evolution. Particularly, cloud tops where phase transition (from liquid to ice) starts and ends needs to be observed by the satellite to profile the thermodynamic phase. The number of these observations has to be significant, since the particle sizes are averaged over a larger domain. So, in general the ensemble method can give an indication when phase transition arises for the first time. However, for estimation of the cloud phase profile at a later stage of the DCC evolution, in situ and also cloud side remote sensing might be the better observation strategy, when phase distribution is altered for example by up- and downdrafts.

*p2,L17: 1) Why is there a new paragraph? 2) Suggest re-wording "In particular ... " as "The phase transition ... is especially relevant for ... " without a preceding indent/new paragraph.*

Changed as suggested:

The phase transition from liquid water to ice is especially is relevant for the development of precipitation.

*p2,L25: remains > remain*

Done.

*P3,L30: "In further development of the scanning ... " Something wrong with the language here and the conclusion of this sentence.*

Changed as follows:

Different from the scanning-point-sensor measurements as presented by Martins et al. (2011), this paper introduces airborne measurements of an imaging spectroradiometer called specMACS (spectrometer of the Munich Aerosol Cloud Scanner, Ewald et al., 2016). These observations were used to derive vertical profiles of the phase state of DCCs during the HALO (High Altitude and Long Range Research Aircraft) campaign ACRIDICON (Aerosol, Cloud, Precipitation, and Radiation Interactions and Dynamics of Convective Cloud Systems) - CHUVA (Cloud processes of tHe main precipitation systems in Brazil: A contribUtion to cloud resolVing modeling and to the GPM (GlobAl Precipitation Measurement)) in 2014 (Wendisch et al., 2016).

*p4,L12: add comma after "September)"*

Done.

p4,L21: The "degree" characters should be superscripts.

Done.

*p5,L7: "by measuring monochromatic radiation from a monochromator" - revise language?*

Changed as follows:

The spectral characteristics were deduced by using monochromator output at selected wavelengths.

p8,l17: "inlay" > "inset"?

Done.

*p9,l31 (top of page) " grid cell in" > "grid cell at"*

Done.

*p9,l2: ranging between > from ... to?*

Changed as suggested.

*p9,l3: What is the "first cloud case"? At this point, the table suggested above would really be helpful.*

The first and the second cloud case are not related with the measured cloud cases which will be presented later in the manuscript. To omit confusion, we called the two cases in this section "cloud scenarios" and modified the introduction of the two scenarios as follows:

Two simplified cloud scenarios with different profiles of cloud effective radius and water content are assumed. In both cases the clouds ranged from 4.0 to 11.0 km altitude with a mixed phase layer between 6.4 and 7.0 km. While the first scenario uses constant values of cloud effective radius ($r_{eff}$ = 20 µm for liquid water and ice) and water content (0.7 gm$^{-3}$), the second scenario assumes variable profiles of the microphysical parameters. These two cases are chosen to identify effects on the $I_P$-profile caused by changes of (i) the phase state itself (scenario 1), and (ii) the cloud particle size and water content (scenario 2).

*p9,l6: "originated" > "originating"?*

Done.

*p9,l8: The phase index is significantly shifted to positive values > either it assumes positive values or not - what is the meaning of "significantly shifted to positive values" How about "shifted to positive values"?*

We rephrased the sentences as follows:

In the mixed phase layer the phase index shows a steep increase to values larger than 0.15. The absolute difference of the phase indices between mixed phase layer and pure ice phase layer is less pronounced than between liquid and mixed phase layer.

*p9,l9: Why "obviously"? Perhaps "apparently"? Meaning unclear.*

We modified it:

This might be caused by the fact that the contribution of ice particles within the mixed phase layer leads to an increased absorption of radiation resulting in an increase of the phase index.

*p9,l14: "related" > "relative"?*

Changed.

*p9,l15: move "also" to after "is"*

Done.

*p9,l20-21: Too hard to understand. Try to improve language.*

Changed as follows:

As concluded in Jäkel et al. (2013), the phase index becomes less variable for a water content of more than 0.4 g m$^{-3}$ (variation lower than 7 %). This holds true for most of the DCCs when cloud edges are excluded, which are optically thinner than the inner regions of the cloud.

*p9,l22: "as can be concluded" > how can this be concluded?*

We combined the two sentences as follows:

Less impact is attributed to the change of the sensor elevation angle, since the variability of the phase index with respect to the viewing geometry for each phase state in the first cloud scenario

with fixed cloud microphysics is lower than the variability of $I_P$ due to the changed cloud properties in the second cloud scenario.

*p10,l1: "showing" > "with"*

Done.

*p10,l2: "need to be taken" > move to right after "images" on l1.*

Changed as follows:

To estimate the distance to the observed cloud element (C) two images from different positions (P1 and P2) with a projection of the observed point in both images need to be taken (C1 and C2, so-called tie points) as illustrated in Fig. 5a

*p10,l2: explain "epipolar plane"*

For easier understanding we removed the two sentences mentioning the "epipolar plane". For the following equations this term is not needed.

*p10,l3: What's the "world" coordinate system?*

A world coordinate system is independent of the camera and aircraft coordinate systems. In fact, the spatial location of the camera/aircraft is given in the world reference system (world coordinate system) with :"The x -and y-axis of the world coordinate system (not shown) are pointed to the east and to the north, respectively, while the z-axis is perpendicular to the x-y plane (pointing upward)." as was stated in the manuscript.

We changed the sentence as follows and added another reference:

The geometric problem comprises three coordinate systems: for the camera, the aircraft, and the world coordinate system (longitude, latitude and altitude) for the observed point C (Biter et al., 1983).

*p10,l6: Usage of the word "exemplarily" seems out of place throughout most of the manuscript. How about "for example"?*

Changed as suggested:

For example, a positive pitch angle of the aircraft …

Here "exemplarily" was removed from the sentence:

The theoretical background on photogrammetry is given in Hartley and Zisserman (2004), while Hu et al. (2009) applied these techniques for cloud geometrical reconstruction.

In Section 3.1 we changed the sentence as follows:

The procedure is applied for an example cloud scene observed during ACRIDICON-CHUVA from 19 September 2014.

In Section 4.1 we exchanged "exemplarily" by "for example":

For example, a closer look at the asphericity is taken for the time range between 18.28 and 18.34 UTC (Fig. 9e).

*p10,l6-7: Unclear what this means - wrong axis perhaps?*

The sentence is extended as follows:

For example, a positive pitch angle of the aircraft (associated with rotation around the aircraft $y_a$-axis) rotates the camera (image) around the camera's $x_i$-axis as can be deduced from Fig. 5b.

*p11,L25: 6b: are the distances in km?*

We added the unit in the text:

From stereographic analysis of these tie points the distances to the cloud points (in km) are determined (Fig. 6b).

*p11,L27: "quite *a* homogeneous" (add "a")*

Done.

*p11,l14: "scientific" > "science"*

Done.

*p11,l16: "which" > "because it" ?*

Changed as suggested.

*p12,L30 (top): "have been" > "were"*

Done.

*p12,l32 (top): "phase states. Mainly" > "phase states, mainly"*

Done.

*p15,L15: ATTO - introduce acronym somewhere.*

The Amazon Tall Tower Observatory – was already introduced in Section 2.1 Field campaign.

---

## Author Comment (AC3) · 9 Jun 2017

**Reply to Reviewer #3:**

We thank the reviewer for the time and efforts she/he spent reading our manuscript and providing valuable advices. Please find below a discussion of the reviewer's comments (italic). Changes/additions made to the text are underlined and given in quotes.

*P4, L14: Is humidity variation small for selected 14 flight cases? If not, influence on the conclusions of this paper should be discussed.*

In another publication of the ACRIDICON-CHUVA special issue, Cecchini et al. (2017b) found differences in the cloud base altitudes which are related to the humidity. Deforestation plays a role to explain contrasts between flights in different Amazonian regions. They found less relative humidity (75 %) and a higher cloud base (2000 m) in the southern region (AC13) which is affected by deforestation compared to measurements over the forest in the north (80 % RH and 1500 m cloud base). This results in a 500 m thinner warm layer for the polluted cloud. Since we didn't measure the humidity at cloud base during the entire time frame of the specMACS cloud side observations, it is difficult to a relation between humidity and cloud evolution in this work. Therefore, we refer to the paper of Cecchini et al. (2017b) where other cloud cases also from other flight days where discussed.
We added the following:

The temperature profiles of the three flights show only small day-to-day variations in spite of the different flight directions. In contrast, the relative humidity is variable with flight area and altitude as was shown by Cecchini et al. (2017b). They discussed in particular the relation between cloud base and humidity below clouds for several flights performed during the ACRIDICON-CHUVA campaign. For AC13 they found less relative humidity (75 %) and a higher cloud base (2000 m) due to deforestation than compared to measurements over the rain forest (80 % relative humidity and 1500 m cloud base).

*Subsection 2.2.2 (MODIS) should be moved to after Subsection 2.2.4 (CAS-DPOL...) because Subsections 2.2.1, 2.2.3 and 2.2.4 describe the aircraft measurements.*

The original intention of the order was to separate remote sensing and in situ instruments. However, we changed the order of chapters as suggested by the reviewer.

*P5, L25: The MODIS thermodynamic phase algorithm should be explained in more detail for better discussion (around P12, bottom) on the comparison of aircraft measurement results with the MODIS phase results.*

We introduced the retrieval of the cloud top phase as provided by the MYD06 data set in more detail and gave some more references. Now it is clearly stated, that with the latest Collection 6, the "mixed phase" class is now combined with the "uncertain" class, in order that a separation of cloud tops with only mixed phase cannot given.

Since MODIS mainly measures cloud top properties, the timespaceexchangeability of convective clouds as proposed by Rosenfeld and Lensky (1998) is applied and referred to as ensemble method. The cloud particle phase of the cloud tops is directly taken from the MOD06/MYD06 20 product "Cloud_Phase_Infrared" with a 1-km-pixel resolution (Baum et al., 2012). Compared to Collection 5, where the cloud phase product was classified as ice, liquid water, mixed phase, and uncertain using brightness temperatures measured at 8.5

and 11 μm (Platnick et al., 2003), Collection 6 is modified by using additional cloud emissivity ratios (7.3/11, 8.5/11, and 11/12 μm) as reported by Pavolonis (2010) and Baum et al. (2012). Empirically derived thresholds of these emissivity ratios were defined to separate finally between liquid water and ice clouds. Note, that due to ambiguity reasons (see Platnick et al. (2017)) a separate classification of mixed phase cloud pixels is no longer provided in Collection 6. The "mixed phase" and "uncertain" classes from Collection 5 are now combined into a single class specified as "undetermined". Hence, the description of the cloud phase profile by applying the ensemble method on the "Cloud_Phase_Infrared" product is limited to the liquid water and the ice phase distribution.
Therefore, the cloud particle size product is used additionally to estimate the glaciation temperature as proposed by Yuan et al. (2010).

The description of the phase statistics as derived from the cloud top and phase frequency plot was modified as follows:

In Fig. 9f the frequency of liquid and ice phase observations for altitude bins of about 200 m is presented. Fully developed deep convective clouds with cloud tops between 10 and 14 km (classified as ice cloud) and low level cumulus clouds up to 6 km (liquid water clouds) are detected. Cloud phase information from the assumed phase transition layers is not available in Collection 6.

And later:

The MODIS phase product shows ice cloud tops between 11 and 15 km altitude and liquid water clouds up to 4.5 km.

*P6, L24, "The aspherical fraction is the ratio of aspherical particles ... ": Is this a ratio of number concentration? Other definitions (area, volume, or mass) of the ratio are possible. Please clarify the definition.*

We specified the definition as follows:

The aspherical fraction (AF) is determined by a size dependent ratio of the polarized backward scattered and the forward scattered light with respect to their number concentration.

*P14, L4, the last sentence of this Subsection, "Also strong downdrafts can ... ": The last part of this sentence, "whereas in situ measurements inside the cloud only reveal liquid phase particles", is confusing to me. Why do cloud side observation and in situ measurements show different results?*

As in situ and cloud side measurements are not collocated in time and space one of the main reasons is the horizontal variability of cloud properties which causes differences in the determination of the cloud phase. This variability is enhanced for increasing vertical velocity within the clouds. We modified the paragraph as follows:

Also local strong downdrafts can transport ice particles into lower levels, which will be interpreted as mixed phase layer from the cloud side observation perspective. Due to the horizontal variability of cloud phase inside a cloud cluster for example caused by up- and downdrafts, in situ measurements may only reveal liquid phase particles. A direct

comparison between the observation strategies is subject to restrictions because of temporal and spatial variability of cloud properties in convective systems.

Typographic corrections

*P9, L32, "140 x 40 x 99": The "x" character should be replaced by the times symbol.*

Done. Also exchanged at other places.

*P9, L20, "get": Should be replaced by "become" or something.*

Done.

*P12, L22, "m s−1": Make the "−1" superscript.*

Done.